# Fusion of crayfish optimization algorithm and MNS-YOLO for solar cell defect detection

Jiayue Zhang[1,2]*, Xinxin Yi[2], Heng Wang[3,4]

**1** School of computer science, Guizhou Police College, Guiyang, Guizhou, China, **2** School of Big Data and Information Engineering, Guizhou University, Guiyang, Guizhou, China, **3** Tongren Polytechnic College, Tongren, Guizhou, China, **4** Guizhou University, Guiyang, Guizhou, China

* zhangjiayue_gzpc@sina.com

## Abstract

Inspection and diagnosis of construction projects involves health monitoring of buildings and related facilities, and the utilization of renewable energy sources, such as solar energy, is critical to the smooth operation of modern construction projects. The detection of solar cell defects is related to the reliability and efficiency of building photovoltaics and has become an area of interest. Existing deep learning-based solar cell defect detection models significantly improve the accuracy of solar cell defect detection, however, deep learning-based solar cell defect detection models ignore the effect of network hyperparameters on their model performance. In this study, the hybrid model CMNS-YOLO, which combines the crawfish optimization algorithm with the MNS-YOLO model, is proposed to achieve the ultimate detection accuracy. First, Mamba-Like Linear Attention is introduced to design the C2f-MLLA module to improve the target feature representation capability of solar cell sheet defects; second, Bidirectional feature pyramid frequency aware feature fusion network is designed to enhance the recovery ability of target detail features as well as the fusion ability of image features; then ShapeIoU is used to solve the target aspect ratio misalignment problem and construct the improved MNS-YOLO network; finally, COA is utilized to adjust the parameters of the MNS-YOLO network. Experimental results on the PV-Multi-Defect and PVELAD datasets show that compared with the baseline model, the detection accuracy of the proposed model on the two datasets is improved by 6.3% and 2.3% while maintaining the lightweight characteristics of the model. Therefore, the proposed method has considerable potential in the field of solar cell defect detection.

## Introduction

With the continuous development of solar energy technology, solar cells have become a widely used clean energy. However, with the continuous expansion of production scale, defects such as scratches and damage that may occur during the

**Data availability statement:** All relevant data are within the manuscript and its Supporting Information files.

**Funding:** This work was supported by Natural Science Research Program for Higher Education Institutions of Guizhou Provincial Department of Education (Youth Science and Technology Talent Growth Program), (Qianjiaoji Support [2024] 183). The funders had no role in study design, data collection and analysis, decision to publish, or preparation of the manuscript.

**Competing interests:** The authors have declared that no competing interests exist.

manufacturing process of solar cells are becoming more and more serious [1]. These defects not only affect the performance and life of the cells, but may also cause equipment failure and safety hazards. Therefore, how to quickly and accurately detect solar cell defects has become an important issue to ensure the production quality of solar cells and improve energy utilization.

Early solar cell defect detection methods mainly rely on manual visual inspection, which requires operation and maintenance engineers to carry instruments to check solar cells one by one. This method is labor-intensive and inefficient, and overly dependent on the subjective experience of operation and maintenance engineers. The detection accuracy cannot be guaranteed. Subsequently, many researchers and scholars have used physical detection methods such as sound waves, lasers, and visible light to detect defects in solar cells [2–7]. In addition, physical detection methods are only suitable for online diagnosis during the wafer and cell manufacturing stages, and have significant limitations in practical applications [8]. Subsequently, traditional computer vision methods based on manual feature extraction and classifiers have been widely used. This method mainly relies on texture, color, and shape feature vectors, and uses classifiers to detect defects. It can achieve non-contact and high-accuracy detection [9]. However, traditional computer vision methods mainly rely on manually extracted descriptors, which require a lot of parameter adjustments during use, and have poor robustness and generalization capabilities. In recent years, deep learning models represented by convolutional neural networks have been widely used in target detection, image classification, semantic segmentation and other fields. Compared with traditional computer vision, deep learning obtains specific feature representations of data sets by learning a large number of samples, and has the advantages of strong robustness, high accuracy, and good generalization ability [10]. At present, deep learning-based detection methods have been applied by many scholars in the field of solar cell defect detection, mainly divided into two directions: precision and lightweight network. In terms of precision, different network structures are designed and different attention mechanisms are integrated to improve detection accuracy [11–13]; in terms of lightweight networks, operations such as deep separable convolution, lightweight network modules, and partial convolution are used to keep the network lightweight while maintaining accuracy [14–16]. Although these networks are effective in identifying specific defects in battery cells, they are not able to extract fine-grained defect features from the battery cells due to insufficient data samples, high similarity of defect features, and complex background features. In addition, feature loss is prone to occur when the network is deepened, resulting in low detection accuracy.

Hyperparameters have a great impact on the performance of deep learning algorithms, and their values vary depending on the specific problem of the algorithm [17]. Therefore, they need to be optimized for different data sets [18]. Hyperparameters based on manual search can be manually estimated through iterative trial and error. This approach can be very challenging for users without sufficient professional background and practical experience. To overcome the shortcomings of manual search, automatic search algorithms such as grid search and random search have been

proposed. Kisvari et al. [19] applied grid search to adjust the hyperparameters of GRU and LSTM models. This method automatically handles the optimization process, but for more complex models, it loses efficiency by increasing the number of hyperparameters and expanding the range of their values [20]. Bergstra et al. [21] showed that random search has the same advantages as grid search, but is more efficient than grid search in high-dimensional space. Nevertheless, in the random selection of combinations, pre-generated hyperparameter combinations cannot meet the needs of specific tasks, especially in complex models [22]. In addition, sequential model-based optimization is an effective algorithm for solving high-dimensional space function optimization problems [23]. In sequential search, several hyperparameters are selected, and after evaluating their quality, the next sampling location is determined. Masum et al. [18] used Bayesian optimization to find the best hyperparameters for deep neural networks when detecting network intrusions, improving model performance. On the other hand, when fitting deep learning models under various hyperparameters, the best hyperparameter setting is crucial, and automatic parameter optimization is the cornerstone of saving time and performance. Erkan et al. [24] proposed a method for identifying plant species from leaf images. This method uses a hyperparameter optimization technique based on the ABC algorithm to optimize the CNN architecture. The ABC algorithm can find the optimal value for some simple structures of the CNN architecture, thereby improving classification performance. Jocher et al. [25] combined hyperparameters with data augmentation techniques to improve target detection accuracy. Karaman [26,27] first integrated the artificial bee colony algorithm into the YOLOv5 algorithm to optimize the hyperparameters of YOLOv5 and improve the accuracy of polyp detection. Nguyen [28] proposed a hybrid SHO-YOLOv5 model, which used the SHO algorithm to optimize the hyperparameters of YOLOv5 and improved the detection effect of the model. However, the effectiveness of the model will decrease when the object is unclear, too small or blurred, and the background is complex.

Existing deep learning-based solar defect detection methods focus on the design of the network structure of the model, while ignoring the impact of the hyperparameters of the model itself on its model performance. Compared with the existing solar cell defect detection methods, the main contributions of this study are as follows:

(1) Introducing Mamba-Like Linear Attention and designing the C2f-MLLA module in combination with C2f to improve the feature representation of the target;

(2) Introducing ShapeIoU to solve the problem of target aspect ratio disproportion in solar cell defect detection dataset;

(3) Innovative design of a bidirectional feature pyramid frequency-aware feature fusion network, which improves the feature fusion capability of the Neck network and the ability to recover target edge details;

(4) In this paper, an improved solar cell chip defect detection method based on MNS-YOLO and crayfish optimization algorithm, i.e., CMNS-YOLO, is proposed, which further improves the performance of the model and can be used for solar cell chip defect detection as well as other detection applications, and provides a reference for future research in this field.

## Related work

### Detection methods based on YOLO and its variants

YOLO has been widely used in the field of computer vision due to its strong real-time performance and fast inference speed [29–31]. Recently, Tian et al. [32] first proposed an attention-centered YOLO framework, namely YOLOv12, which fully utilizes the performance advantages of the attention mechanism while maintaining a speed comparable to the previous CNN-based model. However, YOLOv12 is limited to local information aggregation and pairwise correlation modeling, and lacks the ability to capture global many-to-many high-order correlations. For this reason, Lei et al. [33] seamlessly combined hypergraph computing with end-to-end information collaboration to provide a more accurate, powerful and efficient real-time detection solution YOLOv13, but the inference speed is slightly delayed compared to YOLOv12. In addition

to the traditional YOLO series models, many researchers have further explored the in-depth research of the basic YOLO series models for different scenarios. Huang et al. [34] proposed an end-to-end surface defect detection network SSA-YOLO through a convolution squeeze and excitation module, a Conv2d-BatchNorm-SiLU with a Swin transformer module, and an adaptive spatial feature fusion detection head module, but the detection speed is limited. Chao et al. [35] proposed a metal surface defect detection network IAMF-YOLO based on YOLOv8 information enhancement and multi-scale feature fusion to improve detection accuracy, but the number of model parameters can still be optimized. Recently, Ma et al. [36] designed a linear attention backbone network, a selective feature pyramid network, and a lightweight detection head based on YOLOv8 and proposed a new detection method ELA-YOLO for steel surface defect detection in industry. It achieved a balance between speed and accuracy, but it was difficult to detect a limited number of defect samples and was sensitive to parameter settings.

## Solar cell defect detection method based on machine vision

Traditional machine learning methods [37] manually extract the surface defect features of the cell, and then use traditional algorithms to classify the image texture features. Su et al. [9] used the adjacent pixel information of the original image to calculate the center pixel gradient information, and introduced Center Pixel Gradient Information to Center-Symmetric Local Binary Pattern to extract the gradient features in the EL image. At the same time, similarity analysis and clustering methods were used to obtain the global features of the image, and finally the SVM classifier was used to classify the three types of images: cracks, broken grids, and normal samples. Compared with traditional local binary pattern features, it effectively improves the detection effect of crack defects of different shapes, but cannot express the surface defect characteristics of repeated patterns; Juan et al. [38] proposed the detection technology of electroluminescence imaging combined with support vector machines classifier, which analyzes the impact of defective solar cells on the output efficiency of photovoltaic systems from multiple angles by identifying defects such as defect characteristics, luminescence characteristics and health status of solar cells; Demirci et al. [39] combined the SVM classifier with Deep Feature Based, which can effectively complete the EL image classification task on an imbalanced data set, but there are fewer labeled images in the data set, so the accuracy of the system classification needs to be further improved. Venkatesh et al. [40] used CNN to extract image features, and used the J48 decision tree algorithm to select the most important features from the image features, and used the K nearest neighbor algorithm to perform fault detection on the selected image features. However, this method has high time complexity and will increase the false detection rate when the sample size is unbalanced. Chen et al. [41] designed a solar cell module defect detection method, which performs perspective transformation on tilted solar cell modules, and then directly detects the defects or cuts out individual cells to further classify them into defect categories, and uses EL images to train machine learning models to achieve defect detection. However, this method can only detect the defect distribution and cannot locate the defect. Although traditional machine learning methods have the advantages of good recognition effect and mature algorithms in cell surface defect detection, there are still problems such as the algorithm requiring a large number of parameter adjustments, poor model robustness, poor model generalization performance, reliance on engineers' subjective experience judgment, and inability to perform long-term manual operations.

## Solar cell defect detection method based on deep learning

The cell defect recognition technology of deep learning method [42] includes defect classification and defect detection. For single-category defects in cells, some researchers use defect classification technology of infrared images and electroluminescence images. Zhang et al. [43] fused the Faster R-CNN and R-FCN models and used the improved model to detect surface defects of solar cells. This model makes full use of the small target features extracted by the deep convolutional neural network and effectively improves the accuracy of small defect detection in solar cells, but this model cannot measure the defect size. Su et al. [13] used the electroluminescence (EL) method to automatically detect defects in solar cells. The EL method can detect the existence of defects, but its ability to accurately locate the specific location of the

defects is limited. Et et al. [44] combined VGG-16 and SVM algorithms to propose an efficient hybrid machine learning model to detect whether there are defects in photovoltaic cells. Compared with recent related literature, the model has the highest classification accuracy on the ELPV dataset, but there is still a problem that defects located in the corners of the cell cannot be correctly classified by the model. Zhao et al. [45] used SeFNet to replace the classification layer in HRNet and proposed a solar cell EL image defect detection model called SeF-HRNet, which effectively improved the detection performance of the model. However, the lack of image data in the test set affected the recognition accuracy. Dwivedi et al. [46] used the ViT model based on the attention mechanism for multi-class image classification to identify surface defects from images of solar panels. However, when the model was used for small sample data, its defect detection effect was seriously affected.

## Methods

### YOLOv8 algorithm

YOLOv8 is an end-to-end target detection algorithm proposed by Ultralytics in 2023. It mainly consists of four parts: Input, Backbone, Neck and Head. Among them, the key functions of the Input part include Mosaic data enhancement, adaptive anchor frame computation and adaptive gray scale filling. The Backbone part adopts the feature extraction network of CSPDarkNet idea, which is mainly composed of two parts, C2f and SPPF. The C2f module draws on the ELAN idea of YOLOv7 [47], which connects the gradient flow branches through more to enrich the gradient flow of the model, while the SPPF module is used to adaptively convert feature maps of arbitrary sizes to fixed sizes while realizing the fusion of local and global feature information. The Neck part combines the Feature Pyramid Network [48] and the Path Aggregation Network [49] network architectures, where FPN fuses high-level features with low resolution and high semantic information with low-level features with high resolution and low semantic information to produce rich semantic features at all scales. In contrast, PAN proposes to use adaptive feature pooling to accelerate the flow of information between the bottom and top layers, emphasizing the underlying information and enhancing the overall feature representation. Compared to YOLOv5 [50], the Head part firstly transitions from a coupled head to the contemporary mainstream decoupled head structure, and secondly, shifts from an anchor-frame based paradigm to an anchor-frame free paradigm, which prioritizes the improvement of detection efficiency while streamlining the network structure.

### MNS-YOLO

Facing the serious background interference, low pixel of small targets, and sample aspect ratio disproportion in the targets in the solar panel dataset, which makes the original YOLOv8 have the problems of leakage and misdetection, and low detection accuracy. For solar cell defect detection, the following improvements are made:

(1) Introducing Mamba-Like Linear Attention (MLLA) [51] module and designing C2f-MLLA module, so that the network effectively captures the target location information and local deviation to enhance the feature representation capability of the network;

(2) Designing a bidirectional feature pyramid frequency-aware feature fusion network to enhance the recovery of the original detailed features as well as the fusion of image features;

(3) Introducing ShapeIoU [52] as a loss function for model bounding box regression to solve the problem of severe imbalance in target aspect ratio and improve the accuracy and efficiency of target detection task. The structure of the improved network model MNS-YOLO is shown in Fig 1.

**C2f Meets Mamba-Like Linear Attention.** In the original design of YOLOv8, the C2f module was proposed and applied in its network architecture in order to improve the network performance through cross-stage feature fusion, but it may not be effective in small targets and complex tasks, leading to redundancy or loss of information, which affects the

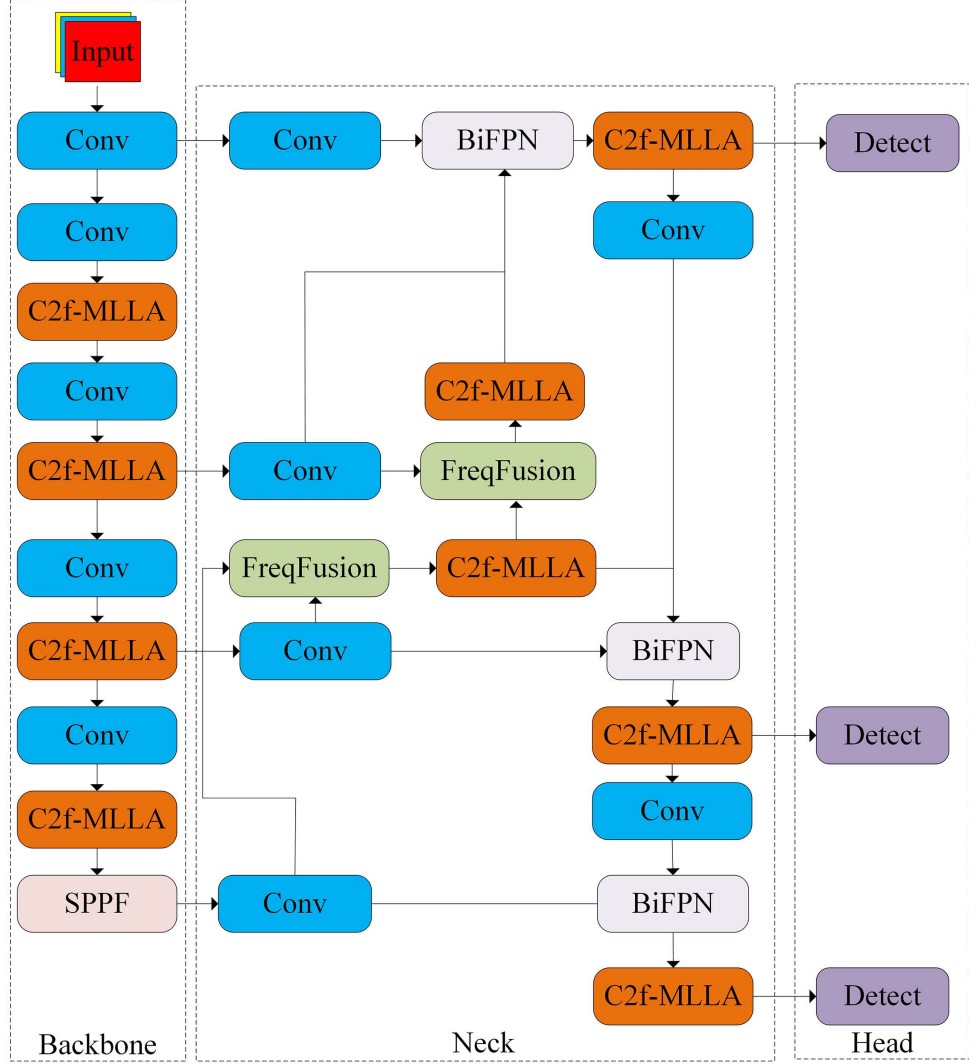

**Fig 1. Structure of MNS-YOLO.**

model performance; moreover, the Bottleneck structure is used in the C2f module to reduce the number of computations and parameters, but the resulting information compression, expression limitations, and gradient vanishing problems may affect the overall performance of the network. Especially in complex tasks or small-objective tasks, excessive compression may lead to performance bottlenecks; therefore, further optimization of the Bottleneck design is needed to reduce the computational overhead while maintaining the expressive ability and training stability of the network. Therefore, this paper combines the C2f and MLLA modules in order to improve the model's performance for solar panel defect detection.

MLLA mainly consists of two key elements, the forget gate and the block design in Mamba, while MLLA provides the necessary positional information by using positional encoding instead of the forget gate, thus maintaining parallel computation and fast inference speed. This makes MLLA more effective in dealing with non-autoregressive vision tasks. Its schematic structure is shown in Fig 2(a).

While fully preserving the advantages of the C2f module's cross-stage feature fusion architecture, this paper replaces the original C2f's Bottleneck structure with the MLLA module (as shown in Fig 2(b)). This design leverages the MLLA

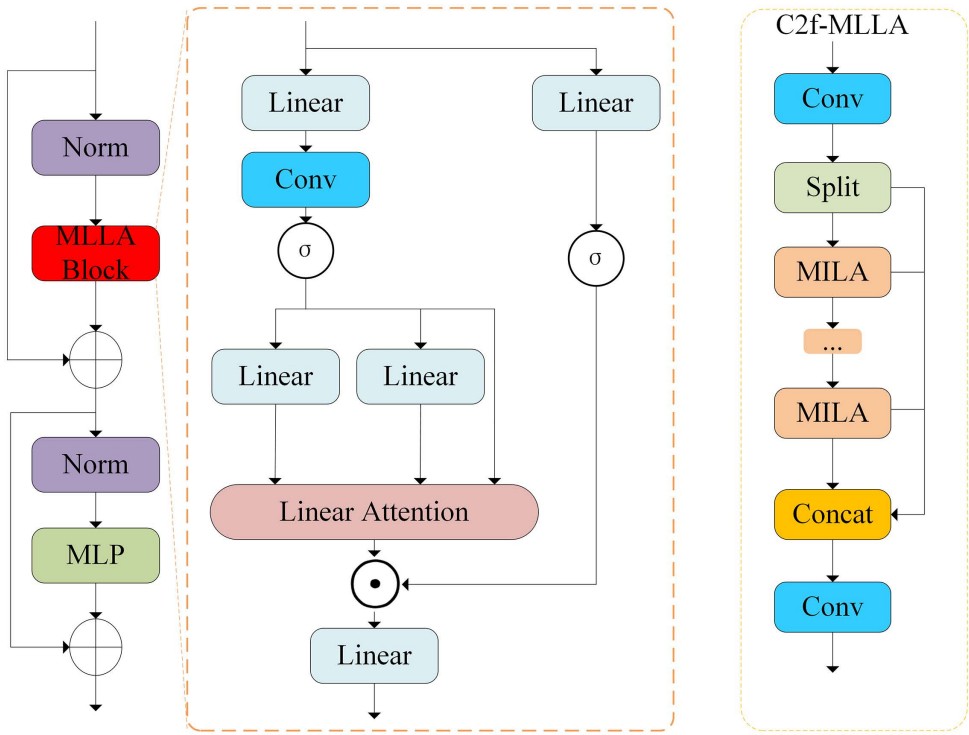

**Fig 2. MLLA module and C2f-MLLA module structure diagram. (a) MLLA module, (b) C2f-MLLA module.**

module's built-in positional encoding mechanism to enable the network to accurately capture spatial position information and local feature deviations when processing image data. This significantly improves the model's sensitivity to the order of input feature sequences, providing critical support for the precise localization of small objects (such as tiny defects in solar panels). Furthermore, the MLLA module avoids the information loss associated with feature compression in traditional Bottleneck architectures by optimizing the information transmission path. This significantly enhances the richness of feature representation while maintaining parallel computing efficiency and fast inference. This structural improvement effectively alleviates performance bottlenecks caused by information redundancy or loss in small object detection tasks, while also enhancing the stability of network training by optimizing gradient flow characteristics.

**Bidirectional feature pyramid frequency aware feature fusion network.** In the design of YOLOv8, the Neck part is mainly responsible for extracting features and fusing them to enhance the detection performance. However, when dealing with complex tasks or small target detection, the detailed features in the lower layers of the Neck module may be masked by the coarse features in the higher layers during multi-layer feature fusion, making it difficult for the network to effectively focus on the key features of the small target. Therefore, in order to solve the problem of small targets in solar panel defect detection, this paper innovatively designs an improved Neck network, i.e., a bidirectional feature pyramid frequency-aware feature fusion network, in order to improve the problems of category consistency and boundary ambiguity in feature fusion. Its structure is shown in Fig 4.

The Frequency-aware Feature Fusion (FFF) module [53] integrates an adaptive low-pass filter (ALPF) generator, an offset generator, and an adaptive high-pass filter (AHPF) generator. The ALPF generator predicts spatially-variable low-pass filters to attenuate high-frequency components within the target, thereby reducing intra-categorical inconsistency; the offset generator optimizes a large range of inconsistent features and fine boundaries by resampling and replacing inconsistent features with more consistent ones, and the AHPF generator enhances the high-frequency boundary detail

information that is lost during downsampling. Its structure is shown in Fig 3(a). And bidirectional feature pyramid net-work(BiFPN) [54] is an efficient feature fusion network for multi-scale feature extraction in target detection. It excels in improving detection accuracy and speed, and is especially suitable for small and multi-scale target detection tasks; it introduces top-down and bottom-up bidirectional connectivity, which allows the information to propagate bi-directionally between different resolution levels, enhances the contextual propagation of features, and improves the detection perfor-mance of the algorithm. Its structure is shown in Fig 3(b).

In this paper, a new Neck network BiFPN-FFFN is designed by combining the FFF module, BIFPN, C2f-MLLA, and convolution module. Its structure is shown in Fig 4. Through the modified design, firstly, the convolution is used for downs-ampling operation to retain the spatial information, secondly, the FFF module is introduced in order to improve the intra-class consistency and enhance the expression of the edge detail features, in addition, the BIFPN is used to effectively fuse different levels of feature information through the top-down and bottom-up feature fusion, and finally, the deviation of feature information is corrected by the C2f-MLLA module in order that the detection head can more fully utilize its feature information to achieve optimal detection.

**ShapeIoU.** For the category problem of severe imbalance of aspect ratio in the dataset, the aspect information of the target frame is incorporated into the final loss calculation to replace the original generic target detection regression loss function, and the loss information is added to the gradient optimization.

The loss function CIoU of the original YOLOv8 model is defined by the ratio of the distance between the centroids of the predicted frame (Anchor frame) and the real frame (GT frame) to the diagonal distance of the minimum closure region as the distance loss, and the aspect ratio between the predicted frame and the real frame as the shape loss.

$$L_{CIoU} = 1 - IoU + \frac{\rho^2\left(b, b^{gt}\right)}{W_g^2 + H_g^2} + \alpha v \tag{1}$$

$$v = 4 * \frac{\left(arctan\left(\frac{w^{gt}}{h^{gt}}\right) - arctan\left(\frac{w}{h}\right)\right)^2}{\pi^2} \tag{2}$$

$$\alpha = \frac{v}{1 - IoU + v} \tag{3}$$

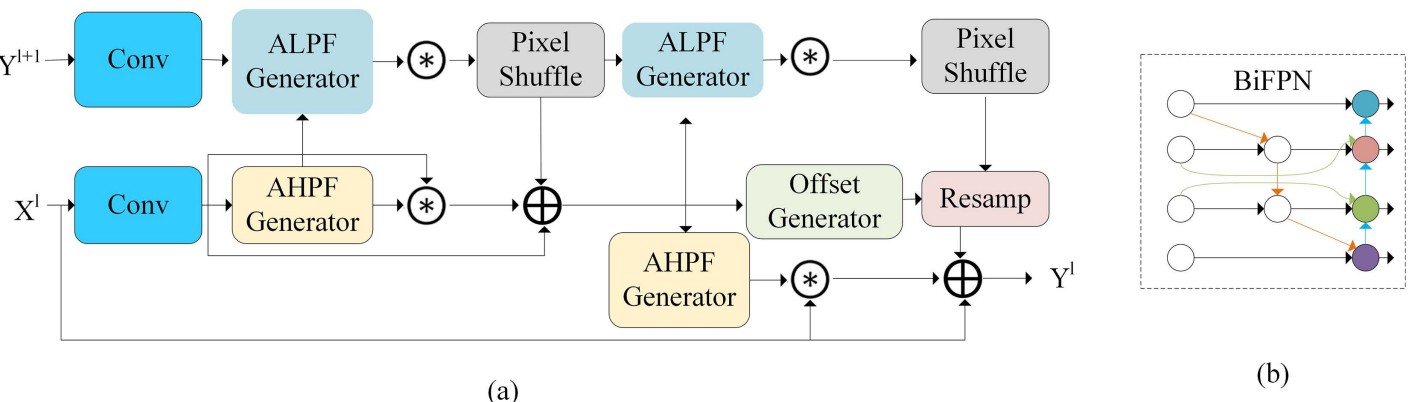

(a)

(b)

**Fig 3. Frequency-aware feature fusion module and BiFPN structure diagram. (a) Frequency-aware Feature Fusion, (b) BiFPN.**

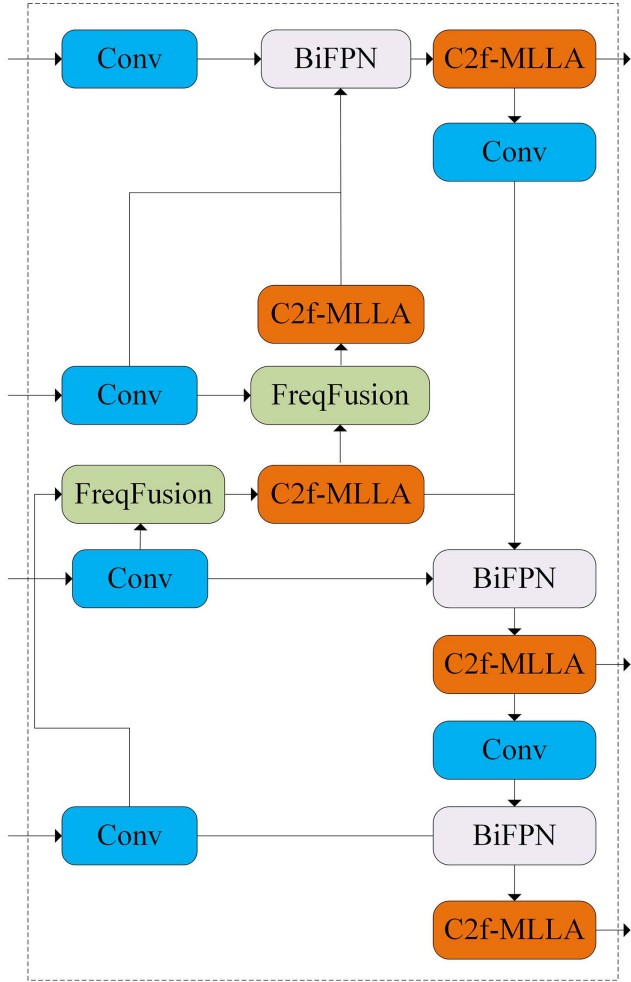

**Fig 4. Bidirectional feature pyramid frequency aware feature fusion network.**

Where *IoU* is the ratio of the intersection-over-union area of the predicted bounding box to the ground-truth bounding box; b is the center point of the predicted bounding box, and $b^{gt}$ is the center point of the ground-truth bounding box; $\rho$ represents the Euclidean distance between the two points. *a* is a weighting factor used to balance the contributions of *IoU* and *v*; *v* is used to calculate the consistency of the aspect ratio between the predicted and target boxes, reflecting the actual difference in height and width, respectively, and their confidence. w, h are the width and height of the predicted box; $w^{gt}$, $h^{gt}$ are the width and height of the ground-truth box. However, when there is an inclusion relationship between the prediction box and the real box, it is guaranteed that the ratio of the width-to-height ratio and the center distance of the prediction box and the real box to the diagonal distance of the minimum enclosing region are the same, and an equal scale enlargement of the smaller boxes in the prediction box and the real box will result in the same value of the CIoU but an increase of the IoU. It indicates that there is a defined shape and distance loss of CoU that is less accurate. Therefore, this paper introduces the ShapeIoU [52] loss function to replace the CIoU loss function in the YOLOv8 model. Its Sha-peIoU does this by focusing on the shape and size information of the border itself and incorporating this information into the IoU loss function, as shown in Eq. 4.

$$L_{ShapeIoU} = 1 - IoU + L_d + 0.5L_\Omega \tag{4}$$

$$L_d = hh \times \left(\frac{x_c - x_c^{gt}}{c}\right)^2 + ww \times \left(\frac{y_c - y_c^{gt}}{c}\right)^2 \tag{5}$$

$$L_\Omega = \sum_{t=w,h} \left(1 - e^{-\omega_t}\right)^\theta, \begin{cases} \omega_w = hh \times \frac{|w - w^{gt}|}{max(w, w^{gt})} \\ \omega_h = ww \times \frac{|h - h^{gt}|}{max(h, h^{gt})} \end{cases} \tag{6}$$

$$ww = \frac{2 \times (w^{gt})^{scale}}{(w^{gt})^{scale} + (h^{gt})^{scale}} \tag{7}$$

$$hh = \frac{2 \times (h^{gt})^{scale}}{(w^{gt})^{scale} + (h^{gt})^{scale}} \tag{8}$$

Among them, $L_d$ is the position deviation loss, $L_\Omega$ is the shape loss, $ww$ and $hh$ are the weight coefficients in the horizontal and vertical directions of the prediction box, respectively, xc and yc are the center coordinates of the bounding box, $x_c^{gt}$ and $y_c^{gt}$ are the center coordinates of the real bounding box, $c$ is the diagonal length of the minimum circumscribed rectangle of the real box and the predicted box, $\omega_w$ and $\omega_h$ are weights related to width and height, and *scale* is the scaling factor, which is related to the size of the target in the dataset, and $\theta$ is the factor of the degree of normalization of the shape loss, which is taken to be 4 in this paper. it can be seen that ShapeIoU does the splitting of the distance loss in the x and y directions, and can calculate the distance loss in the x and y directions separately, optimizing the CIoU distance loss is not defined accurately. The introduction of horizontal and vertical weight coefficients enhances adaptability to irregularly shaped objects and reduces bounding box positioning bias.

### Crayfish optimization algorithm

Crayfish Optimization Algorithm (COA) was proposed by Jia et al. [55] in 2023. COA was inspired by the crayfish foraging, heat avoidance and competition behaviors, which were divided into three different phases to balance the exploration phase and development phase of the algorithm. Its three phases are heat avoidance phase, competition phase and foraging phase, where heat avoidance phase represents the exploration phase of COA and competition and foraging phases represent the development phase of COA.

**Initializing populations.** In a multidimensional optimization problem, each crayfish is a $1 \times dim$ matrix, and each crayfish represents a solution to the problem in a set of variables ($X_{i,1}, X_{i,2}, ..., X_{i,dim}$), and each variable $X_i$ must be located between the upper and lower boundaries of the variables, and the initialization of the COA is to generate a set of candidate solutions X randomly in a space with the initialization of COA is shown in Eq. 9.

$$X = [X_1, X_2, \ldots, X_i, \ldots, X_N]^T, X_i = [X_{i,1}, \ldots, X_{i,j}, \ldots, X_{i,dim}] \tag{9}$$

where $X$ is the initial population position, $N$ is the population number, $dim$ is the population dimension, and $X_{i,j}$ is the position of individual $i$ in dimension $j$, as calculated by Eq. 10.

$$X_{i,j} = rand\left(ub_j - lb_j\right) + lb_j \tag{10}$$

where $lb_j$ denotes the lower bound of the $j$th dimension, $ub_j$ denotes the upper bound of the $j$th dimension, and rand is a random number from Ezugwu et al. [56].

**Definition of temperature and crayfish intake.** Crayfish behavior is affected by changes in temperature. When the temperature is higher than 30°C, crayfish will choose a shady place to avoid the heat. At suitable temperatures, crayfish will engage in foraging behavior. Therefore, the COA defines a temperature range of 20–35°C. The mathematical models of ambient temperature and crayfish intake are shown in Eq. 11 and Eq. 12.

$$temp = rand \times 15 + 20 \tag{11}$$

$$p = C_1 \times \left( \frac{1}{\sqrt{2\pi}\sigma} \times exp\left( -\left( \frac{temp - \mu}{2\sigma^2} \right)^2 \right) \right) \tag{12}$$

where $temp$ represents the temperature of the environment where the crayfish are located, $\mu$ is the temperature best suited for the growth of crayfish, and $\sigma$ and $C_1$ are used to control the intake of crayfish $p$ at different temperatures.

**Summer escape stage.** When $temp > 30$, the crayfish will enter the burrow to escape the heat. The burrow $\boldsymbol{X}_{shade}$ is defined as follows:

$$\boldsymbol{X}_{shade} = \frac{\boldsymbol{X}_G + \boldsymbol{X}_L}{2} \tag{13}$$

Where $\boldsymbol{X}_G$ denotes the optimal position obtained so far through the number of iterations and $\boldsymbol{X}_L$ denotes the optimal position of the current population.

When $rand < 0.5$, it means that there are no other crayfish competing for the burrow, at which time the crayfish utilize Eq. 14 for summering.

$$\boldsymbol{X}_{i,j}^{t+1} = \boldsymbol{X}_{i,j}^t + C_2 \times rand\left( \boldsymbol{X}_{shade} - \boldsymbol{X}_{i,j}^t \right) \tag{14}$$

$$C_2 = 2 - \frac{t}{T} \tag{15}$$

where $C_2$ is the decay curve, $t$ denotes the current number of iterations, and $T$ denotes the maximum number of iterations.

**Competition stage.** When $temp > 30$ and $rand \geq 0.5$, crayfish compete for burrows via Eq. 16.

$$\boldsymbol{X}_{i,j}^{t+1} = \boldsymbol{X}_{i,j}^t - \boldsymbol{X}_{z,j}^t + \boldsymbol{X}_{shade} \tag{16}$$

where $z$ represents a random individual of crayfish as shown in Eq. 17.

$$z = round\left( rand \times (N-1) \right) + 1 \tag{17}$$

**Foraging stage.** When $temp \leq 30°C$, crayfish engage in foraging behavior with food location $\boldsymbol{X}_{food}$ and food size $Q$ defined as Eq. 18 and Eq. 19:

$$\boldsymbol{X}_{food} = \boldsymbol{X}_G \tag{18}$$

$$Q = C_3 \times rand \times \frac{fitness_i}{fitness_{food}} \tag{19}$$

Where $C_3$ is the food factor, representing the largest food, the value is constant 3, *fitness$_i$* represents the fitness value of the ith crayfish, and *fitness$_{food}$* represents the fitness value of the food.

When $Q > (C_3 + 1)/2$, it means that the food is too big, and the crayfish use its claws to tear the food, as shown in Eq. 20:

$$\boldsymbol{X}_{food} = exp\left(-\frac{1}{Q}\right) \times \boldsymbol{X}_{food} \tag{20}$$

After the food is shredded by the crayfish, the crayfish forage based on the intake as shown in Eq. 21:

$$\boldsymbol{X}_{i,j}^{t+1} = \boldsymbol{X}_{i,j}^{t} + \boldsymbol{X}_{food} \times p \times (cos(2\pi \times rand) - sin(2\pi \times rand)) \tag{21}$$

When $Q \leq (C_3 + 1)/2$, crayfish move toward food and forage as shown in Eq. 22:

$$\boldsymbol{X}_{i,j}^{t+1} = \left(\boldsymbol{X}_{i,j}^{t} - \boldsymbol{X}_{food}\right) \times p + p \times rand \times \boldsymbol{X}_{i,j}^{t} \tag{22}$$

## CMNS-YOLO

In Section 3.2, this paper proposes the MNS-YOLO algorithm, although MNS-YOLO makes many improvements relative to the original YOLOv8, MNS-YOLO uses hyperparameters obtained by training on the MS-COCO dataset, and the default hyperparameters often do not achieve the best results when facing different datasets or application scenarios, and it is usually necessary to adjust the hyperparameters empirically or manually based on the values. However, there are 30 commonly used hyperparameters in MNS-YOLO, and the existence of many hyperparameters makes determining the optimal hyperparameter combination a complex task. Exhaustive search of high-dimensional hyperparameter space is both time-consuming and costly. With the in-depth study of swarm intelligence mechanisms and deep learning models, Karaman provides a new paradigm for hyperparameter optimization of deep learning models [27].

In this paper, we will use the COA algorithm to optimize the hyperparameters in the MNS-YOLO algorithm. Its core lies in achieving a global hyperparameter search through three stages of behavioral simulation. First, the summer-avoidance behavior of crayfish is used to explore the search range. Second, resource competition among crayfish randomly perturbs the position to enhance local search accuracy. Finally, the crayfish's sense of smell allows for directional movement and precise foraging. This multi-stage collaborative mechanism can dynamically balance the breadth and depth of the hyperparameter search space, dynamically adjust each hyperparameter configuration, and avoid parameter sensitivity, so as to obtain the optimal hyperparameter combination and prevent MNS-YOLO from encountering performance bottlenecks. The structure of its COA optimized MNS-YOLO is shown in Fig 5.

As can be seen from Fig 5, the hyperparameters of COA optimization MNS-YOLO are mainly composed of three parts: dataset and data preprocessing, CMNS-YOLO. The steps of the algorithm are as follows, first, the solar panel defect dataset is collected, and in order to prevent network training from overfitting, data enhancement techniques such as random rotation, random cropping, and introduction of noise are used to increase the diversity of the images in the dataset so as to improve the performance and generalization ability of the model. Secondly, before using the COA algorithm for MNS-YOLO hyperparameter optimization, the hyperparameters that need to be optimized are initialized (the same as the original MNS-YOLO hyperparameters), and in this paper, 22 common hyperparameters in MNS-YOLO are considered, such as the initial learning rate, the bounding-box loss gain, and the initial momentum of the warm-up, as shown in Table 1.

After initializing the hyperparameters of MNS-YOLO, we redefined the population initialization of COA, which is defined as shown in Eq. 23.

$$\boldsymbol{X} = [\boldsymbol{X}_1, \boldsymbol{X}_2, \ldots, \boldsymbol{X}_i, \ldots, \boldsymbol{X}_N]^T, \boldsymbol{X}_i = \left[lr0_i, lrf_i, \ldots, mixup_i, copy_{paste_i}\right] \tag{23}$$

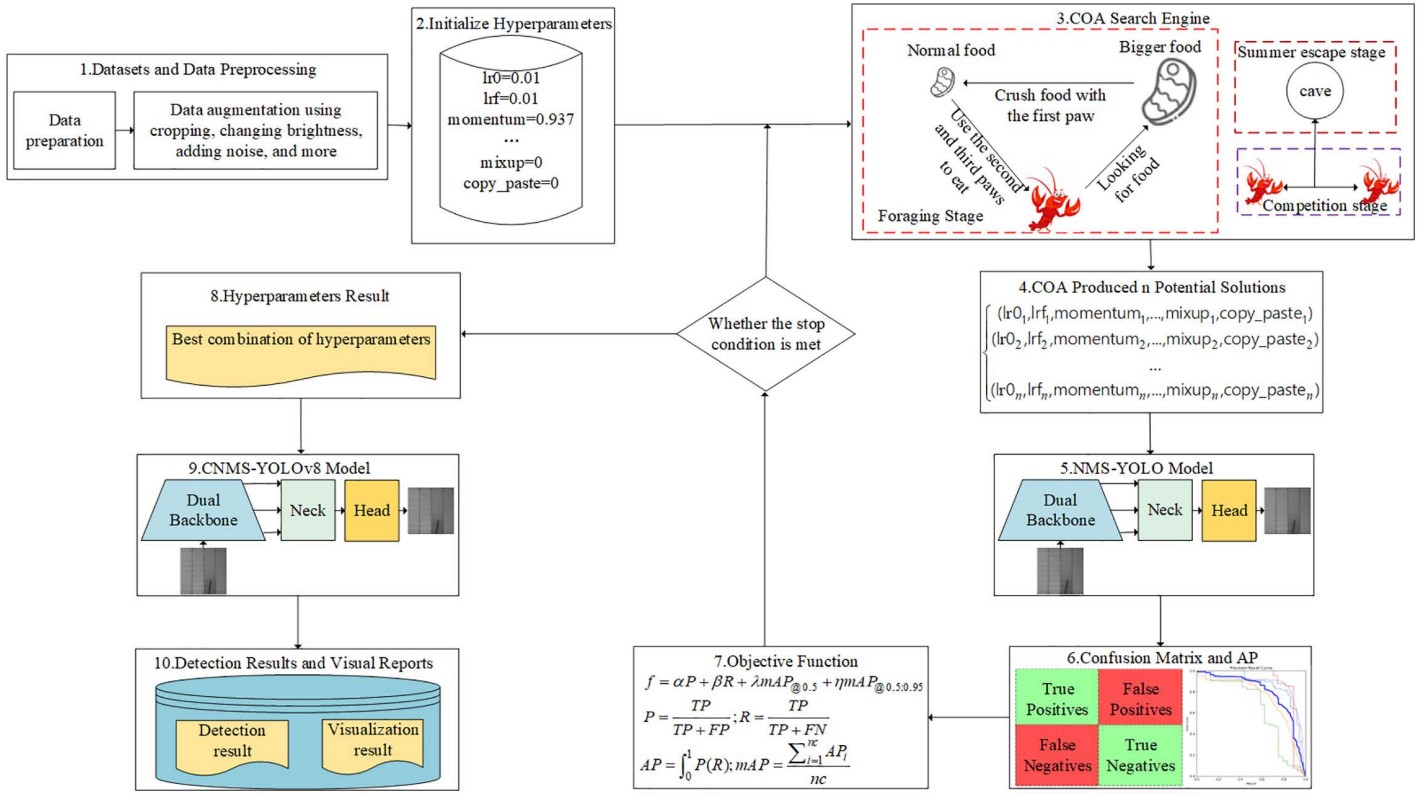

**Fig 5. General structure of COA optimized MNS-YOLO.**

Where $N$ denotes the number of populations, $lr0_i, lrf_i, \ldots, mixup_i, copy_{paste_i}$ are the hyper-parameters of MNS-YOLO, whose values are randomly initialized using Eq. 23 according to the bounding ranges in Table 1, and in this paper, we limit the hyper-parameters to be initially 0 not to change with the change of iterations.

Usually, for the target detection algorithm, its performance metrics are shown in Section 4.3, therefore, in this paper, P, R, and mAP@0.5 and mAP@0.5:0.95 are chosen as the fitness function to optimize the MNS-YOLO algorithm. Its adaptation degree is defined as shown in Eq. 24.

$$fitness = \alpha \times P + \beta \times R + \lambda \times mAP@0.5 + \mu \times mAP@0.5:0.95 \tag{24}$$

where $\alpha$, $\gamma$, $\beta$ and $\mu$ are constants belonging to the range between [0,1]. In this paper, $\alpha = \beta\ \mu = 0$, $\lambda = 1$. The COA algorithm aims to find the best solution by using as a fitness function to optimize the hyperparameters of the MNS-YOLO network. After obtaining the best hyperparameter combination by optimizing MNS-YOLO through COA, we again initialize the hyperparameters of MNS-YOLO by using the best hyperparameter combination for the MNS-YOLO model, and finally fine-tune it on the dataset to obtain the CMNS-YOLO model, and analyze and evaluate the performance of its model.

## Experiments

### Experimental environment and parameter settings

All experiments in this paper were conducted under Ubuntu 20.04, where the CPU was Intel(R)Xeon(R)Platinum-8352V@2.10GHz, the memory was 90GB, the GPU was NVIDIAGeForceRTX4090, the video memory was 24GB, the

**Table 1. Hyperparameter settings.**

| Description | Notation | Boundary value |
|---|---|---|
| Initiallearningrate | lr0 | [1.00E-05,1.00E-01] |
| Finallearningrate | lrf | [0.01,1] |
| Adammomentum | momentum | [0.6,0.98] |
| Optimizerweightdecay | weight_decay | [0.0,0.001] |
| Warmupepochs | warmup_epochs | [0.0,5] |
| Warmupinitialmomentum | warmup_momentum | [0.0,0.95] |
| Boxlossgain | box | [0.0,10] |
| Clslossgain | cls | [0.0,10] |
| Dfllossgain | dfl | [0.0,10] |
| ImageHSV-Hueaugmentation | hsv_h | [0.0,0.1] |
| ImageHSV-Saturationaugmentation | hsv_s | [0.0,0.9] |
| ImageHSV-Valueaugmentation | hsv_v | [0.0,0.9] |
| Imagerotation | degrees | [0.0,45.0] |
| Imagetranslation | translate | [0.0,0.9] |
| Imagescale | scale | [0.0,0.9] |
| Imageshear | shear | [0.0,10.0] |
| Imageperspective | perspective | [0.0,0.001] |
| Imageflipup-downprobability | flipud | [0.0,1.0] |
| Imageflipleft-rightprobability | fliplr | [0.0,1.0] |
| Imagemixupprobability | mosaic | [0.0,1.0] |
| Imagemixupprobability | mixup | [0.0,1.0] |
| Segmentcopy-pasteprobability | copy_paste | [0.0,1.0] |

development environment used Jupyter Notebook, the programming language was Python 3.8, and the training was accelerated based on Pytorch 1.10.0 using CUDA11.3. Its parameter settings are shown in Table 2.

## Dataset

The PV-Multi-Defect [57] dataset is an open dataset containing five categories, including five types of defects, namely damaged (broken), highlight (hot-spot), black_border, scratch, and no_electricity, as shown in Fig 6, which has a total of 1106 images, whose image size is 600×600, and a total of 3981 target regions are labeled.

Since the PV-Multi-Defect dataset has few defect samples, data enhancement is used in this paper to avoid a series of problems such as overfitting or underfitting of the model during the training process. The dataset is data-enhanced by flipping, rotating, contrast adjustment and adding noise to solve the solar cell sheet defect detection problem, and the final dataset contains a total of 3,318 solar cell sheet defect images. As shown in Fig 7, (a) denotes the original image; (b) denotes flipping in order to simulate the possible defect locations of the solar cell wafers; (c) denotes rotation in order to

**Table 2. Experimental parameter settings.**

| Parameter | Value |
|---|---|
| Epoch | 200 |
| Batch size | 16 |
| Optimizer | Adam |
| COA iterations | 30 |

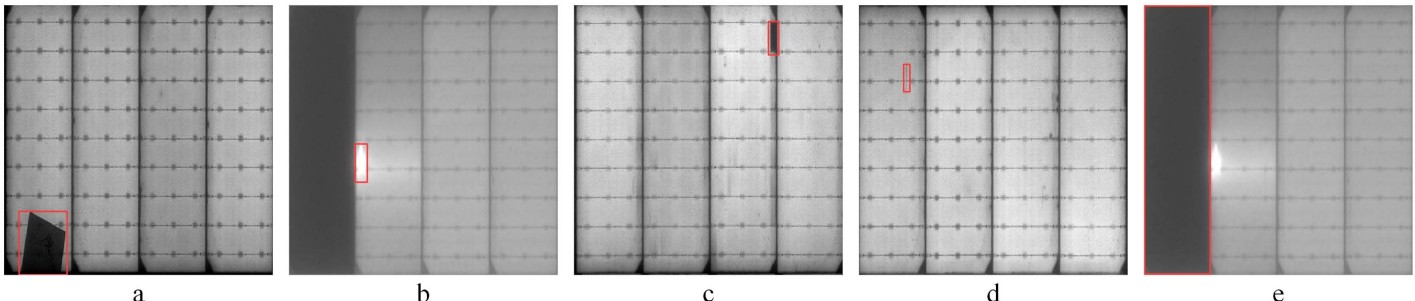

**Fig 6. Dataset defect type.** (a) Cell with broken areas; (b) Cell with obvious bright areas; (c) Cell with black or gray border areas;(d) Scratch cell;(e) No-electricity,showing the black area of the cell.

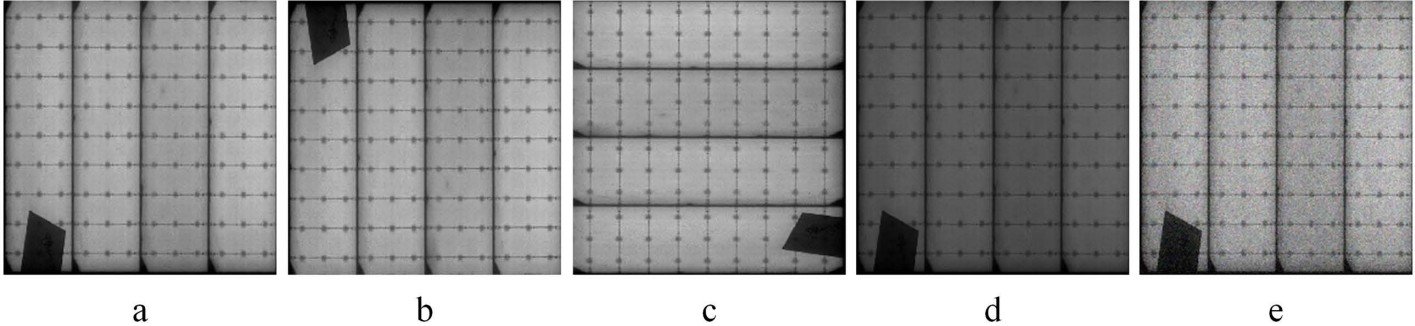

**Fig 7. Example of data enhancement.**

enhance the diversity and complexity of the images; (d) denotes the contrast-brightness adjustment, which addresses the fact that the actual captured image may not be able to highlight the tiny defects that may be present in the image due to the brightness and contrast issues; (e) denotes the noise in so as to enhance the robustness of the model. And the dataset is divided into training set, validation set and test set according to the ratio of 8:1:1.

## Evaluation indicators

In order to evaluate the performance and effectiveness of CMNSYOLO, this paper selects Precision (P), Recall (R), and mean Average Precision (mAP) [58] to evaluate the model. The calculation formula is as follows:

$$P = \frac{TP}{TP+FP} \tag{25}$$

$$R = \frac{TP}{TP+FN} \tag{26}$$

$$AP = \int_0^1 P(R)dR \tag{27}$$

$$mAP = \frac{\sum_{i=1}^{nc} AP_i}{nc} \tag{28}$$

Among them, *TP*, *FP* and *FN* refer to the number of positive samples predicted by the model as positive, the number of negative samples predicted by the model as positive, and the number of positive samples predicted by the model as negative, respectively, and *nc* represents the number of classification categories. P measures the accuracy of the model in predicting positive samples, R represents the number of samples predicted by the model as positive samples, AP represents the area enclosed by the PR curve and the coordinate axis, and mAP represents the average value of AP corresponding to each category. The larger the value, the better the model effect. In order to reflect the performance of the model on the entire dataset, in the mAP calculation, the IoU threshold is taken as 0.5, recorded as mAP@0.5, which means that the detection is considered successful only when the IoU between the real box and the model prediction box is greater than 0.5, and mAP@0.5:0.95 is the mAP calculated at different IoU thresholds (from 0.5–0.95, with a step size of 0.05). In addition, the number of model parameters and GFLOPs [59] are used as performance evaluation criteria for model lightweighting, and the number of frames per second (FPS) is used as an evaluation indicator of the model detection speed to verify whether the model meets the requirements of real-time monitoring.

## Experimental results and analysis

**Ablation experiment.** In order to analyze the extent to which the modification of each module and the combination of modules affects the optimization of the algorithm's performance, a series of ablation experiments were carried out, and a total of 9 experimental scenarios were designed with four improvement modules numbered individually, i.e., A (C2f-MLLA), B (BiFPN-FFFN), C (ShapeIoU), and D (COA), which were based on the YOLOv8 network, and were gradually incorporated into the improved modules. These experiments are trained sequentially in the described experimental environment and the best weight file of each training output is taken on the validation set, and the obtained experimental data are shown in Table 3.

As can be seen from Table 3, the MNS-YOLO algorithm in this paper, after introducing the C2f-MLLA, BiFPN-FFFN and ShapeIoU modules on the basis of the YOLOv8 algorithm, improves the P, R and mAP@0.5 and mAP@0.5:0.95 of the YOLOv8 algorithm by 7.9, 11.4, 4.2 and 3.3 percentage points each compared to the YOLOv8 algorithm, which verifies its module's effectiveness, and that each module alone or in combination performs better than the original YOLOv8 algorithm. Moreover, the model only reduces 0.2 parameters and increases 0.6 GFLOPs. In addition, the model's inference speed FPS only drops by 22.8, but the FPS is still as high as 285.1, which meets the needs of real-time monitoring. Considering that different combinations of hyperparameters will affect the performance of the model, the initial hyperparameter values of the original YOLOv8 model are derived from the MS-COCO dataset, and the hyperparameter values of the solar cell defects dataset may need to be adjusted accordingly, therefore, in this paper, a new CMNS-YOLO model is formed through the combination of the MNS-YOLO and the COA in order to automatically perform the hyperparameter

**Table 3. Ablation experiments. In the table, A represents the C2f-MLLA module, B represents the BiFPN-FFFN module, C represents the ShapeIoU module, and D represents the COA algorithm.**

| A | B | C | D | *P* | *R* | *mAP@0.5* | *mAP@0.5:0.95* | *Parameters* | *GFLOPs* | *FPS* |
|---|---|---|---|---|---|---|---|---|---|---|
| | | | | 0.762 | 0.734 | 0.798 | 0.524 | 3.1 | **8.2** | 307.9 |
| √ | | | | 0.789 | 0.807 | 0.821 | 0.537 | 3.2 | 8.3 | **301.4** |
| | √ | | | 0.772 | 0.792 | 0.817 | 0.529 | **2.8** | 8.7 | 290.7 |
| | | √ | | 0.768 | 0.784 | 0.811 | 0.526 | 3.1 | **8.2** | 307.9 |
| √ | √ | | | 0.813 | 0.826 | 0.831 | 0.534 | 2.9 | 8.8 | 285.1 |
| √ | | √ | | 0.82 | 0.838 | 0.835 | 0.541 | 3.2 | 8.3 | 301.4 |
| | √ | √ | | 0.809 | 0.821 | 0.828 | 0.530 | **2.8** | 8.7 | 290.7 |
| √ | √ | √ | | 0.841 | 0.848 | 0.84 | 0.557 | 2.9 | 8.8 | 285.1 |
| √ | √ | √ | √ | **0.857** | **0.862** | **0.861** | **0.573** | 2.9 | 8.8 | 285.1 |

optimization search. Its performance is improved by 0.4, 0.6, 0.9 and 1.1 percentage points each with respect to the P, R, mAP@0.5 and mAP@0.5:0.95 of MNS-YOLO, And keep the model's parameters, GFLOPs, and FPS consistent with MNS-YOLO, which indicates that the optimization of MNS-YOLO model using meta-heuristic algorithms is effective and feasible, and it can be Better solar cell defect detection tasks can be achieved without increasing the model parameters, GFLOPs and FPS.

**Comparison experiments.** In order to further validate the effectiveness of the algorithm improvement in this paper and verify the improvement of the improved algorithm in terms of P, R, mAP@0.5 and mAP@0.5:0.95, a series of comparison experiments are conducted. Classical and mainstream target detection algorithms are compared on the PV-Multi-Defect dataset, such as YOLOv5 [50], YOLOv6 [51], YOLOv7 [47], YOLOv9-c [53], YOLOv10n [60], YOLOv11n [61], GELAN-c [53], RT-DETER-R50 [62] and the detection effect of the algorithm in this paper, the experimental results are shown in Table 4.

By comparing the experimental results of different algorithmic models in Table 4, it can be clearly observed that the CMNS-YOLO proposed in this paper is significantly better than the other algorithms in terms of P, R, and mAP@0.5 and mAP@0.5:0.95 values, and compared with the YOLOv8n algorithm, it improves its P, R, and mAP@0.5 and mAP@0.5:0.95 by 9.5 each, 12.8, 6.3, and 4.9 percentage points; compared with the state-of-the-art YOLOv11n algorithm, its P, R, and mAP@0.5 and mAP@0.5:0.95 are each improved by 0.9, 0.8, 0.8, and 0.5 percentage points; In addition, in terms of parameter quantity and computational complexity, CMNS-YOLO is almost the same as the baseline model YOLOv8n. Compared with the mainstream lightweight models YOLOv10n and YOLOv11n, its parameter scale is slightly higher but the computational complexity is moderate, which belongs to the lightweight category and is suitable for edge device deployment; although its GFLOPs is slightly higher than YOLOv11n, its FPS is significantly better than YOLOv10n and YOLOv11n, indicating that it has achieved a good balance between computational efficiency and inference speed. From the perspective of accuracy, parameter quantity and floating-point operation, CMNS-YOLO has achieved detection accuracy close to SOTA with extremely small parameter quantity and computational cost; therefore, through comparative experiments, it can be concluded that the CMNS-YOLO detection model can effectively detect defects in solar cells, achieve better detection performance, and achieve a balance between accuracy and efficiency.

**Hyperparameter optimization results and analysis.** In order to verify the effectiveness of the COA algorithm in optimizing the YOLOv8n network and its improved network MNS-YOLO, this paper takes the PV-Multi-Defect dataset as an example, and combines the COA algorithm with YOLOv8n and MNS-YOLO respectively, namely C-YOLO and CMNS-YOLO, and compares them with the original network YOLOv8n and MNS-YOLO network. The experimental results are shown in Table 5. As shown in Table 5, compared with the baseline model YOLOv8n, the P, R, mAP@0.5, and

**Table 4. Model comparison experiment.**

| Model | P | R | mAP@0.5 | mAP@0.5:0.95 | Parameters | GFLOPs | FPS |
|---|---|---|---|---|---|---|---|
| YOLOv5n | 0.761 | 0.684 | 0.741 | 0.465 | **2.5** | 7.2 | 287.2 |
| YOLOv6n | 0.747 | 0.692 | 0.763 | 0.481 | 4.2 | 11.9 | **363.2** |
| YOLOv7-tiny | 0.755 | 0.725 | 0.778 | 0.497 | 6.2 | 13.8 | 286.3 |
| YOLOv8n | 0.762 | 0.734 | 0.798 | 0.524 | 3.1 | 8.2 | 307.9 |
| CELAN-c | 0.833 | 0.781 | 0.846 | 0.548 | 25.3 | 103.1 | 144.6 |
| YOLOv9-c | 0.769 | 0.788 | 0.826 | 0.536 | 25.6 | 103.7 | 142.9 |
| YOLOv10n | 0.845 | 0.816 | 0.853 | 0.555 | 2.7 | 8.4 | 232.1 |
| YOLOv11n | 0.848 | 0.854 | 0.853 | 0.568 | 2.59 | **6.5** | 275.2 |
| RT-DETER-R50 | 0.837 | 0.793 | 0.848 | 0.553 | 42.8 | 130.5 | 39.9 |
| MNS-YOLO | 0.841 | 0.848 | 0.84 | 0.557 | 2.9 | 8.8 | 285.1 |
| CMNS-YOLO(Ours) | **0.857** | **0.862** | **0.861** | **0.573** | 2.9 | 8.8 | 285.1 |

**Table 5. Performance of COA on different models.**

| Model | P | R | mAP@0.5 | mAP@0.5:0.95 | Parameters | GFLOPs | FPS |
|---|---|---|---|---|---|---|---|
| YOLOv8n | 0.762 | 0.734 | 0.798 | 0.524 | 3.1 | 8.2 | 307.9 |
| C-YOLO | 0.821 | 0.819 | 0.827 | 0.543 | 3.1 | 8.2 | 307.9 |
| MNS-YOLO | 0.841 | 0.848 | 0.840 | 0.557 | 2.9 | 8.8 | 285.1 |
| CMNS-YOLO (Ours) | **0.857** | **0.862** | **0.861** | **0.573** | 2.9 | 8.8 | 285.1 |

mAP@0.5:0.95 values of C-YOLO increased by 5.9, 8.5, 2.9, and 1.9 percentage points respectively; compared with the baseline model MNS-YOLO, the P, R, mAP@0.5, and mAP@0.5:0.95 values of CMNS-YOLO increased by 1.6, 1.4, 2.1, and 1.6 percentage points respectively; and the introduction of the COA algorithm does not increase the number of model parameters, floating-point operations, and the inference speed of the detection model. Therefore, the COA algorithm can maintain the original number of model parameters, floating-point operations, and inference speed while improving the detection accuracy of the model.

In addition, this paper takes the PV-Multi-Defect dataset as an example. The initial default hyperparameters of MNS-YOLO are shown in Default in Table 6, and the optimal hyperparameter results of the MNS-YOLO model optimized by COA are shown in Best in Table 6. Fig 8 further illustrates the correlation between the 22 hyperparameters used in the COA algorithm optimization of MNS-YOLO and the performance metrics after 30 training sessions. The vertical axis represents the hyperparameter values and the horizontal axis represents the fitness values associated with each particular hyperparameter. As can be seen in Fig 8, the hyperparameter values tend to cluster around the initial hyperparameter value. The hyperparameters initially set to 0 remain unchanged in each training iteration, while the remaining hyperparameters are modified and recorded after each run. The hyperparameters with the best results produced are denoted by a blue plus sign in the figure, their original hyperparameters are denoted by a red plus sign in the figure, and the color of the dots denotes the density of the points at that location, with brighter colors denoting a tighter grouping of fitness values and hyperparameters at that location.

**Generalization experiment.** In order to verify the generalization ability of the CMNS-YOLO algorithm and the effectiveness of MNS-YOLO, this paper uses the general target detection public dataset PVELAD [63] to verify the generalization of CMNS-YOLO's detection performance. which contains 36,543 near-infrared images with various internal defects and heterogeneous backgrounds. This dataset contains 1 class of anomaly-free images and anomalyous images with 12 different categories such as crack (line and star), finger interruption, black core, thick line, scratch, fragment,

**Table 6. Hyperparameter optimization results.**

| Parameters | Default | Best | Parameters | Default | Best |
|---|---|---|---|---|---|
| lr0 | 0.01 | 0.01 | hsv_v | 0.4 | 0.439 |
| lrf | 0.01 | 0.0103 | degrees | 0.0 | 0.0 |
| momentum | 0.937 | 0.939 | translate | 0.1 | 0.111 |
| weight_decay | 0.0005 | 0.00049 | scale | 0.5 | 0.512 |
| warmup_epochs | 3.0 | 2.81 | shear | 0.0 | 0.0 |
| warmup_momentum | 0.8 | 0.82 | perspective | 0.0 | 0.0 |
| box | 7.5 | 6.41 | flipud | 0.0 | 0.0 |
| cls | 0.5 | 0.428 | fliplr | 0.5 | 0.475 |
| dfl | 1.5 | 1.52 | mosaic | 1.0 | 0.903 |
| hsv_h | 0.015 | 0.0152 | mixup | 0.0 | 0.0 |
| hsv_s | 0.7 | 0.67 | copy_paste | 0.0 | 0.0 |

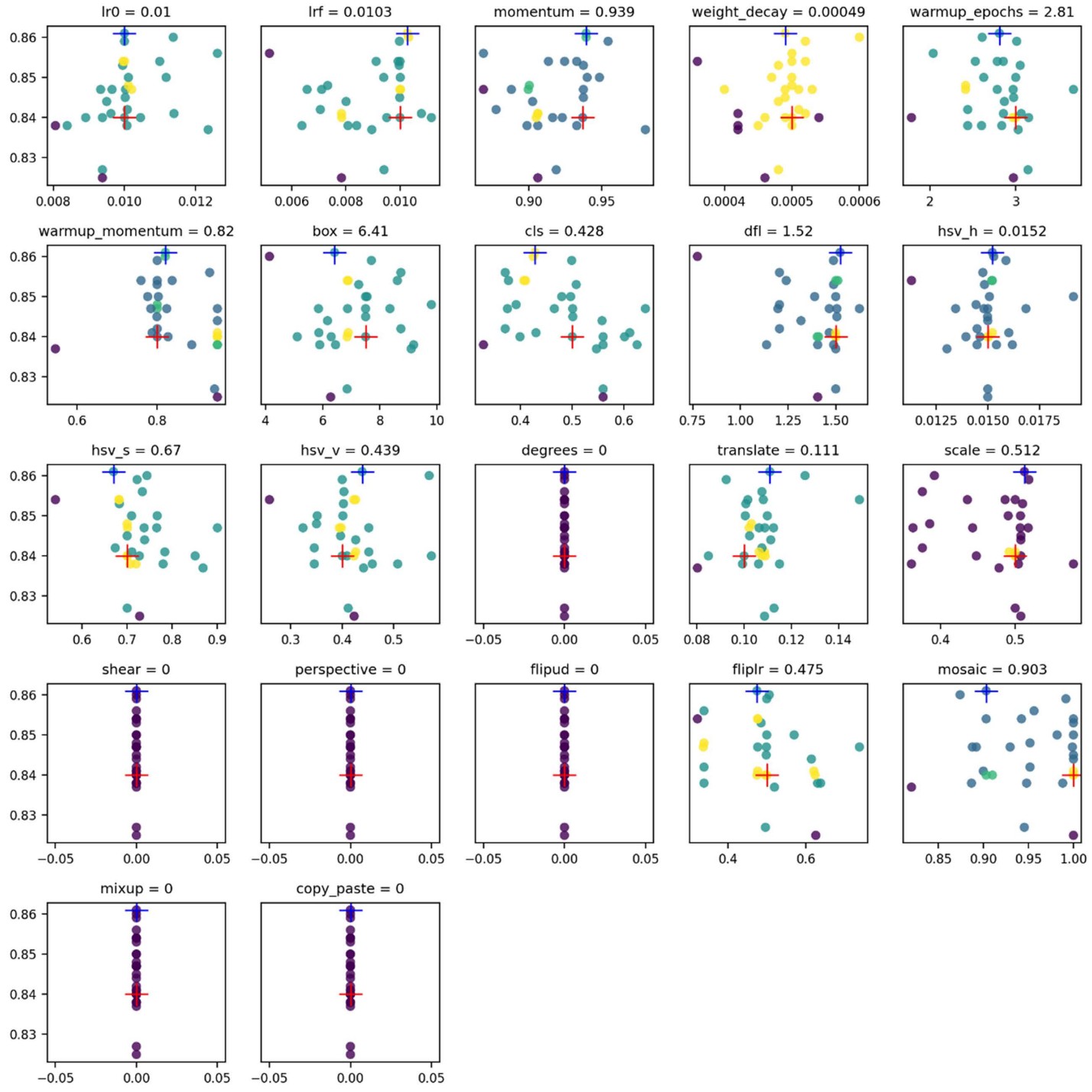

**Fig 8. Visualization of hyperparameter optimization results.**

corner, printing_error, horizontal_dislocation, vertical_dislocation, and short_circuit defects. Moreover, 40358 ground truth bounding boxes are provided for 12 types of defects. When the model running environment remains consistent, the experimental results are shown in Table 7.

**Table 7. Comparison of experimental results on PVELAD dataset.**

| Model | P | R | mAP@0.5 | mAP@0.5:0.95 | Parameters | GFLOPs | FPS |
|---|---|---|---|---|---|---|---|
| YOLOv5n | 0.754 | 0.688 | 0.818 | 0.475 | **2.5** | 7.2 | 287.2 |
| YOLOv6n | 0.703 | 0.832 | 0.781 | 0.494 | 4.2 | 11.9 | **363.2** |
| YOLOv7-tiny | 0.852 | 0.757 | 0.820 | 0.532 | 6.2 | 13.8 | 286.3 |
| YOLOv8n | 0.878 | 0.779 | 0.833 | 0.561 | 3.1 | 8.2 | 307.9 |
| CELAN-c | 0.793 | 0.742 | 0.801 | 0.521 | 25.3 | 103.1 | 144.6 |
| YOLOv9-c | 0.657 | 0.834 | 0.767 | 0.507 | 25.6 | 103.7 | 142.9 |
| YOLOv10n | 0.650 | 0.841 | 0.831 | 0.518 | 2.7 | 8.4 | 232.1 |
| YOLOv11n | 0.876 | 0.787 | 0.836 | 0.569 | 2.59 | **6.5** | 275.2 |
| RT-DETER-R50 | 0.866 | 0.769 | 0.823 | 0.554 | 42.8 | 130.5 | 39.9 |
| MNS-YOLO | 0.881 | 0.817 | 0.847 | 0.573 | 2.9 | 8.8 | 285.1 |
| CMNS-YOLO(Ours) | **0.887** | **0.849** | **0.856** | **0.579** | 2.9 | 8.8 | 285.1 |

By comparing the experimental results of different algorithm models in Table 7, it can be clearly observed that the CMNS-YOLO proposed in this paper is significantly better than other algorithms in terms of P, R, mAP@0.5 and mAP@0.5:0.95 values. Compared with the YOLOv8n algorithm, its P, R, mAP@0.5 and mAP@0.5:0.95 are improved by 0.9, 7.0, 2.3 and 1.8 percentage points respectively; compared with the most advanced YOLOv11n algorithm, P, R, mAP@0.5 and mAP@0.5:0.95 are improved by 1.1, 6.2, 1.6 and 1.0 percentage points respectively. In addition, although CMNS-YOLO's FPS is slightly lower than YOLOv8n and YOLOv6n, it is significantly better than other high-precision models, such as RT-DETR-R50, and has the same number of parameters and floating-point calculations as the light-weight model; although YOLOv11n has lower parameters and calculations, its mAP@0.5:0.95 and FPS are slightly lower than CMNS-YOLO. Therefore, it can be observed that CMNS-YOLO performs best in the balance between accuracy and efficiency, can retain its accuracy to the maximum extent, and is suitable for scenarios with high requirements for detection accuracy and real-time performance. This result not only verifies the effectiveness of the proposed algorithm, but also demonstrates the excellent generalization ability of CMNS-YOLO in practical applications.

**Visualization of detection results.** In conclusion, the CMNS-YOLO algorithm proposed in this study has the best detection accuracy and significant overall performance, thus arguing the advantages of the algorithm. In order to evaluate the improvement effect more intuitively, this paper randomly selects different types of solar cell defect images from the PV-Multi-Defect dataset for testing, and the results are shown in Fig 9. In the first and second images of Fig 9(a), the original YOLOv8n algorithm has the problem of leakage detection, such as the first image of (a) cannot detect the black_border, and the second image of (a) cannot detect the small target scratch in the lower right corner, and the other improved algorithms are able to detect all the defects of the images and the confidence level of the CMNS-YOLO algorithm is the highest; For the third and fifth images of Fig 9(a), both YOLOv8n and its improved methods can detect broken, but the CMNS-YOLO algorithm detects it with the highest confidence; For the fourth image in Fig 9(a), YOLOv8n can only detect no_electricity, but MNS-YOLO can detect hot-spot on top of that, while CMNS-YOLO not only detects hot-spot, but also further divides the hot-spot on the left side into two small regions hot-spot, and for the hot-spot region of the right region detects two region-wide hotspots; It can be seen from Fig 9 that the improved CMNS-YOLO does not have the problem of leakage and misdetection, and has a higher confidence level of detection and more accurate localization of the bounding box. In the PV-Multi-Defect dataset detection task, this study improves the YOLOv8n algorithm, which effectively solves the leakage and misdetection problems of the original algorithm, and at the same time significantly improves the average recognition accuracy of the defects on the surface of solar cells. These results show that the CMNS-YOLO algorithm has great potential and practical application value in the field of solar cell wafer defect detection.

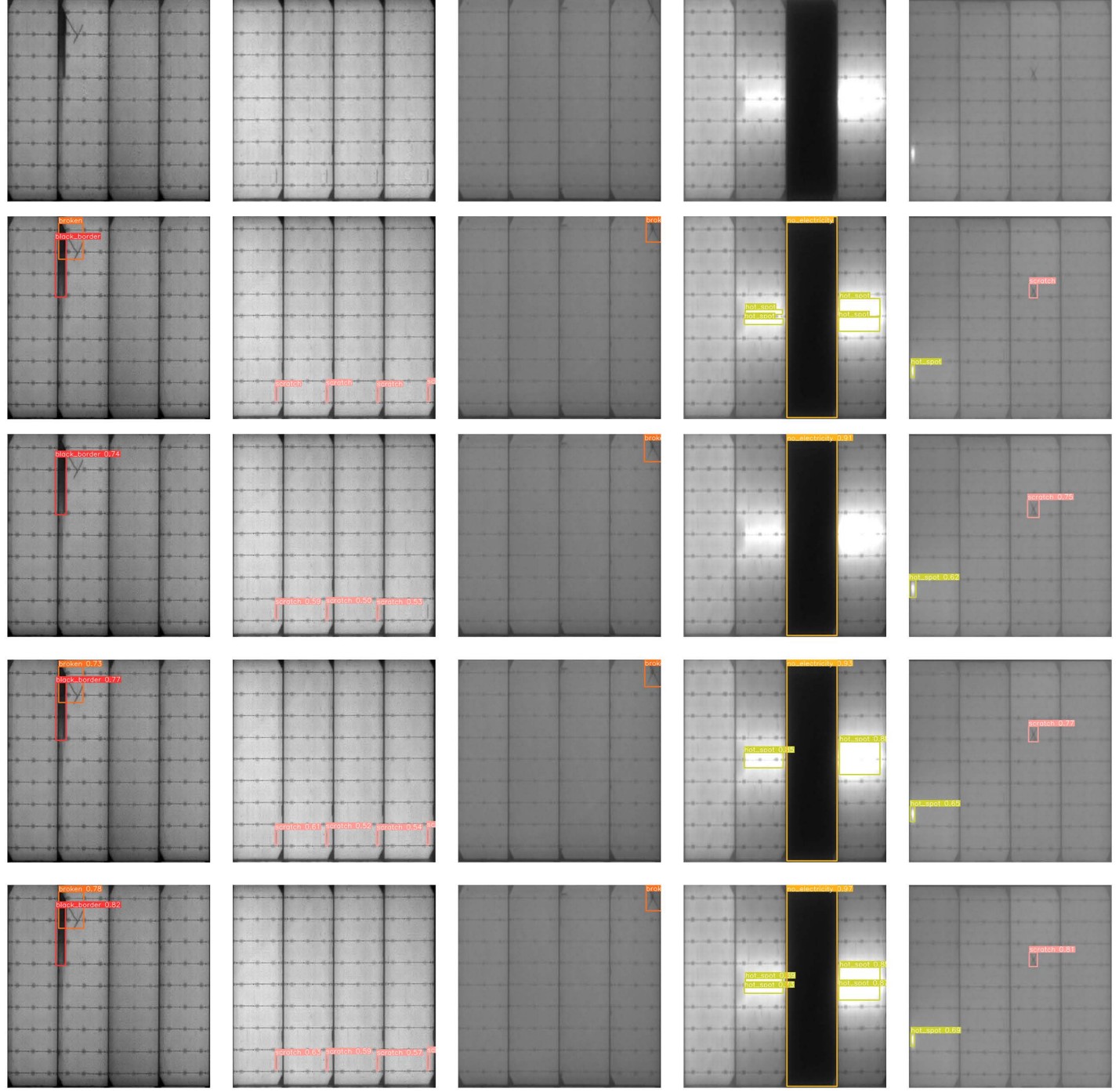

**Fig 9. Comparison of detection effect before and after improvement.** (a) Original figure; (b) real frame; (c) YOLOv8n detection effect; (d) MNS-YOLO detection effect; (e) CMNS-YOLO detection effect.

## Conclusion

Hidden crack defects in solar cell wafers have a huge impact on the panel module, but in a solar cell defect image, the defects have a small percentage of area in the picture. The area characteristic of the cell sheet leads to high difficulty in detecting defects on it. In order to meet this requirement, the CMNS-YOLO model is designed in this study for detecting defects on solar cell wafers. YOLOv8 network firstly, the C2f-MLLA module is designed by introducing the MLLA module to enhance the feature representation capability of the network; secondly, the BiFPN-FFF network is designed to enhance the recovery capability of the original detail features and the fusion capability of the image features; finally, ShapeIoU is introduced as the loss function of the model bounding box regression to solve the target disproportion problem and improve the target accuracy and efficiency of the detection task, and the MNS-YOLO model is obtained through the above modules. Subsequently, a novel and efficient optimization algorithm COA is applied to fine-tune the parameters of MNS-YOLO to obtain the CMNS-YOLO detection model. The solution was developed to cope with the difficulty of detecting defects in different solar cells, and the integration of automated methods helps to overcome the inherent limitations of manual inspection and monitoring methods.

In order to assess the relative effectiveness of the proposed model, experimental validation was performed on the PV-Multi-Defect dataset and further verified the generalization of the model on the PVELAD dataset. It is worth noting that the proposed method requires a higher computational cost compared to other models. However, the empirical findings strongly demonstrate that the optimized CMNS-YOLO is the most effective model for precision and recall metrics. The test results on the test dataset effectively validate the performance of the model, demonstrating its ability to detect with high precision, fewer missed cases, and meet real-time operational requirements. Although the current model exhibits significant advantages, it is not without limitations. Notably, the training time of the model is long, especially when dealing with large datasets. In the future, the training time of the model will be further optimized. In addition, the adaptive design of the model highlights its potential utility in addressing various detection challenges in solar cell defect detection.

## Supporting information

**S1 File. Minimal data set.**
(ZIP)

**S2 File. Values used to build graphs.**
(ZIP)

## Author contributions

**Conceptualization:** Xinxin Yi.

**Methodology:** Jiayue Zhang, Xinxin Yi, Heng Wang.

**Validation:** Xinxin Yi.

**Visualization:** Jiayue Zhang, Heng Wang.

**Writing – original draft:** Jiayue Zhang.

**Writing – review & editing:** Heng Wang.

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
