## [Decision Letter · Decision Letter 0]

13 Feb 2025

PONE-D-25-02298Fusion of crayfish optimization algorithm and MNS-YOLO for solar cell defect detectionPLOS ONE

Dear Dr. zhang,

Thank you for submitting your manuscript to PLOS ONE. After careful consideration, we feel that it has merit but does not fully meet PLOS ONE’s publication criteria as it currently stands. Therefore, we invite you to submit a revised version of the manuscript that addresses the points raised during the review process.

We look forward to receiving your revised manuscript.

Kind regards,

Xu Yanwu

Academic Editor

PLOS ONE

Additional Editor Comments (if provided):

Reviewers' comments:

Reviewer's Responses to Questions

**Comments to the Author**

1. Is the manuscript technically sound, and do the data support the conclusions?

Reviewer #1: Yes

Reviewer #2: Partly

2. Has the statistical analysis been performed appropriately and rigorously? 

Reviewer #1: Yes

Reviewer #2: Yes

3. Have the authors made all data underlying the findings in their manuscript fully available?

Reviewer #1: Yes

Reviewer #2: Yes

4. Is the manuscript presented in an intelligible fashion and written in standard English?

Reviewer #1: Yes

Reviewer #2: Yes

5. Review Comments to the Author

Reviewer #1: The title of this article is "Fusion of crayfish optimization algorithm and MNS-YOLO for solar cell defect detection." It explores a new defect detection method aimed at improving the accuracy and efficiency of solar cell inspection.

Key Points:

1) Background: The article highlights the importance of defect detection in solar cells for enhancing photovoltaic system efficiency. Traditional methods have limitations, necessitating new algorithms to improve detection performance.

2) Methodology: The authors propose a method combining the crayfish optimization algorithm with the MNS-YOLO model, a deep learning-based object detection algorithm effective in identifying and locating various defects in solar cells.

3) Experimental Results: Tests on the PV-Multi-Defect dataset show that the CMNS-YOLO model outperforms other mainstream algorithms (e.g., YOLOv5, YOLOv6) in precision (P), recall (R), and mean average precision (mAP), demonstrating its effectiveness in solar cell defect detection.

4) Data Augmentation: To address dataset limitations, the article uses data augmentation techniques like flipping, rotation, contrast adjustment, and noise addition to enhance model robustness.

Comparison with EFS-YOLO:

EFS-YOLO has advantages in lightweight design, adaptability, resource efficiency, and experimental validation. It reduces model parameters and computational complexity through the EfficientViT backbone, CFE module, and shared parameter detection heads, improving real-time performance and edge device deployment. EFS-YOLO's robustness and versatility are validated on datasets like NEU-DET and GC10-DET, showing a good balance between lightweight design and detection accuracy. In contrast, CMNS-YOLO focuses more on network structure and target accuracy but lacks attention to lightweight design and versatility, limiting its application scope.

Comparison with YOLO-CXR:

"YOLO-CXR: A Novel Detection Network for Locating Multiple Small Lesions in Chest X-Ray Images" excels in several areas compared to the solar cell defect detection article. YOLO-CXR addresses the complex task of detecting multiple small lesions in chest X-rays, offering higher practical value for faster and more accurate diagnosis of chest diseases. It introduces innovations like RefConv layers, the Efficient Channel and Local Attention (ECLA) mechanism, and a dedicated small-lesion detection head, significantly improving feature extraction and small-lesion detection accuracy. Additionally, YOLO-CXR uses the large and diverse VinDr-CXR dataset, with rigorous experimental design and ablation studies, validating its effectiveness. In contrast, the solar cell defect detection article, while proposing improvements to MNS-YOLO, has fewer innovations, a smaller dataset, and simpler experimental design. Overall, YOLO-CXR demonstrates higher academic value and practical potential.

Drawbacks:

1) Long Training Time: The article notes that while the model shows significant advantages, training time is long for large datasets, potentially affecting practical application efficiency.

2) Limited Sample Size: Despite data augmentation, the original dataset has limited defect samples, which may impact the model's generalization ability.

3) Model Complexity: Combining multiple algorithms increases implementation complexity, requiring more computational resources and time in real-world applications.

4) Limitations: The article does not extensively discuss the model's performance under different environmental conditions. Future research could validate the model in more complex real-world scenarios.

In summary, this article provides a new approach to solar cell defect detection but faces challenges in practical application that need to be addressed.

References:

1. Chen B, Wei M, Liu J, et al. EFS-YOLO: a lightweight network based on steel strip surface defect detection[J]. Measurement Science and Technology, 2024, 35(11): 116003.

2. Hao S, Li X, Peng W, et al. YOLO-CXR: A novel detection network for locating multiple small lesions in chest X-ray images[J]. IEEE Access, 2024

Reviewer #2: 1.The dataset is relatively small; it is recommended to use a larger dataset in the future or conduct cross-dataset validation to further verify the model's generalization ability.

2.Compared with several mainstream object detection algorithms, CMNS-YOLO performs well in multiple indicators. However, it is not mentioned whether the hyperparameter settings of the baseline model have been optimized, and it is recommended to add details to ensure the fairness of the comparison experiment.

3.The images in the article are unclear, and furthermore, they should be inserted in the appropriate positions within the article.

6. PLOS authors have the option to publish the peer review history of their article (what does this mean? ). If published, this will include your full peer review and any attached files.

**Do you want your identity to be public for this peer review?** For information about this choice, including consent withdrawal, please see our Privacy Policy .

Reviewer #1: No

Reviewer #2: No

---

## [Author Response · Author response to Decision Letter 1]

7 Apr 2025

Reviewer #1:

The title of this article is "Fusion of crayfish optimization algorithm and MNS-YOLO for solar cell defect detection." It explores a new defect detection method aimed at improving the accuracy and efficiency of solar cell inspection.

Key Points:

1) Background: The article highlights the importance of defect detection in solar cells for enhancing photovoltaic system efficiency. Traditional methods have limitations, necessitating new algorithms to improve detection performance.

2) Methodology: The authors propose a method combining the crayfish optimization algorithm with the MNS-YOLO model, a deep learning-based object detection algorithm effective in identifying and locating various defects in solar cells.

3) Experimental Results: Tests on the PV-Multi-Defect dataset show that the CMNS-YOLO model outperforms other mainstream algorithms (e.g., YOLOv5, YOLOv6) in precision (P), recall (R), and mean average precision (mAP), demonstrating its effectiveness in solar cell defect detection.

4) Data Augmentation: To address dataset limitations, the article uses data augmentation techniques like flipping, rotation, contrast adjustment, and noise addition to enhance model robustness.

Comparison with EFS-YOLO:

EFS-YOLO has advantages in lightweight design, adaptability, resource efficiency, and experimental validation. It reduces model parameters and computational complexity through the EfficientViT backbone, CFE module, and shared parameter detection heads, improving real-time performance and edge device deployment. EFS-YOLO's robustness and versatility are validated on datasets like NEU-DET and GC10-DET, showing a good balance between lightweight design and detection accuracy. In contrast, CMNS-YOLO focuses more on network structure and target accuracy but lacks attention to lightweight design and versatility, limiting its application scope.

Comparison with YOLO-CXR:

"YOLO-CXR: A Novel Detection Network for Locating Multiple Small Lesions in Chest X-Ray Images" excels in several areas compared to the solar cell defect detection article. YOLO-CXR addresses the complex task of detecting multiple small lesions in chest X-rays, offering higher practical value for faster and more accurate diagnosis of chest diseases. It introduces innovations like RefConv layers, the Efficient Channel and Local Attention (ECLA) mechanism, and a dedicated small-lesion detection head, significantly improving feature extraction and small-lesion detection accuracy. Additionally, YOLO-CXR uses the large and diverse VinDr-CXR dataset, with rigorous experimental design and ablation studies, validating its effectiveness. In contrast, the solar cell defect detection article, while proposing improvements to MNS-YOLO, has fewer innovations, a smaller dataset, and simpler experimental design. Overall, YOLO-CXR demonstrates higher academic value and practical potential.

Drawbacks:

1) Long Training Time: The article notes that while the model shows significant advantages, training time is long for large datasets, potentially affecting practical application efficiency.

2) Limited Sample Size: Despite data augmentation, the original dataset has limited defect samples, which may impact the model's generalization ability.

3) Model Complexity: Combining multiple algorithms increases implementation complexity, requiring more computational resources and time in real-world applications.

4) Limitations: The article does not extensively discuss the model's performance under different environmental conditions. Future research could validate the model in more complex real-world scenarios.

In summary, this article provides a new approach to solar cell defect detection but faces challenges in practical application that need to be addressed.

References:

1. Chen B, Wei M, Liu J, et al. EFS-YOLO: a lightweight network based on steel strip surface defect detection[J]. Measurement Science and Technology, 2024, 35(11): 116003.

2. Hao S, Li X, Peng W, et al. YOLO-CXR: A novel detection network for locating multiple small lesions in chest X-ray images[J]. IEEE Access, 2024

Response: Thank you for your professional comments. We have taken your comments into consideration. First, we further validate the generalization ability of the model and the generality of the model on the public dataset of generic target detection, PVELAD (Please see P618, in red); second, we use the number of model parameters, GFLOPs, as a performance evaluation criterion for model lightweight, and use the number of frames per second, FPS, as an evaluation metric for the speed of the model detection in order to verify whether the model meets the requirements of real-time monitoring. (Please see P515,536,568, in red). Experiments are conducted to further verify that the model achieves high detection accuracy and generalization ability while maintaining a lightweight design.

We believe that these modifications, effectively highlight the academic value and practical potential of the CMNS-YOLO model proposed in this paper. We appreciate your suggestion as it helps to enhance the engineering value of this study.

Reviewer #2:

1. The dataset is relatively small; it is recommended to use a larger dataset in the future or conduct cross-dataset validation to further verify the model's generalization ability.

Response:Thank you for your professional comments. We have taken your comments into consideration and further validated the generalization ability of the model on the generalized target detection public dataset PVELAD. (Please see P618, in red).

These modifications, in our opinion, effectively highlight the significance of the CMNS-YOLO model proposed in this paper. We thank you for your suggestion as it helped us to further validate the validity as well as the generalization ability of the proposed model and to improve the clarity and persuasiveness of the results presented in the manuscript.

2.Compared with several mainstream object detection algorithms, CMNS-YOLO performs well in multiple indicators. However, it is not mentioned whether the hyperparameter settings of the baseline model have been optimized, and it is recommended to add details to ensure the fairness of the comparison experiment.

Response:Thank you for your specialized comments. We have considered your comments and combined the COA algorithm with YOLOv8n and MNS-YOLO, i.e., C-YOLO and CMNS-YOLO, respectively, and compared them with the baseline model YOLOv8n and MNS-YOLO networks to further verify the effectiveness of the COA algorithm in optimizing the network hyperparameters, and give the parameter changes before and after the model optimization. (Please see P584, in red).

We hope that this explanation will clarify the effectiveness of the COA algorithm in optimizing the hyperparameters of YOLO or modified YOLO and show the correlation between the values of the initial hyperparameters of the baseline model as well as the optimized hyperparameters and the performance metrics. We appreciate your suggestion as it helps to enhance the academic rigor of this study.

3.The images in the article are unclear, and furthermore, they should be inserted in the appropriate positions within the article.

Response:Thank you for your professional comments. We have taken your comments into consideration replaced the images in the original article with high-resolution images, in addition, and inserted them in the appropriate places in the article. By making these changes, we have improved the quality of the paper. We appreciate your suggestions as they help to improve the readability and quality of the manuscript.

---

## [Decision Letter · Decision Letter 1]

27 Jun 2025

PONE-D-25-02298R1Fusion of crayfish optimization algorithm and MNS-YOLO for solar cell defect detectionPLOS ONE

Dear Dr. zhang,

Thank you for submitting your manuscript to PLOS ONE. After careful consideration, we feel that it has merit but does not fully meet PLOS ONE’s publication criteria as it currently stands. Therefore, we invite you to submit a revised version of the manuscript that addresses the points raised during the review process.

We look forward to receiving your revised manuscript.

Kind regards,

Xu Yanwu

Academic Editor

PLOS ONE

Journal Requirements:

Additional Editor Comments:

Please address the issues from Reviewer 4. Note that extra experiments are not necessary.

Reviewers' comments:

Reviewer's Responses to Questions

**Comments to the Author**

1. If the authors have adequately addressed your comments raised in a previous round of review and you feel that this manuscript is now acceptable for publication, you may indicate that here to bypass the “Comments to the Author” section, enter your conflict of interest statement in the “Confidential to Editor” section, and submit your "Accept" recommendation.

Reviewer #2: All comments have been addressed

Reviewer #3: All comments have been addressed

Reviewer #4: (No Response)

2. Is the manuscript technically sound, and do the data support the conclusions?

Reviewer #2: Yes

Reviewer #3: Yes

Reviewer #4: Partly

3. Has the statistical analysis been performed appropriately and rigorously? 

Reviewer #2: Yes

Reviewer #3: Yes

Reviewer #4: N/A

4. Have the authors made all data underlying the findings in their manuscript fully available?

Reviewer #2: Yes

Reviewer #3: Yes

Reviewer #4: No

5. Is the manuscript presented in an intelligible fashion and written in standard English?

Reviewer #2: Yes

Reviewer #3: Yes

Reviewer #4: No

6. Review Comments to the Author

Reviewer #2: (No Response)

Reviewer #3: Authors had adderessed all the concerns suggested by the reviewers, I have no question for publication.

Reviewer #4: The manuscript proposes CMNS-YOLO, a model combining the crawfisher optimization algorithm with MNS-YOLO for solar cell defect detection. However, critical flaws in motivation, theoretical foundation, and experimental design undermine the validity of the research. Some suggestions are as follows:

Motivation: Recent YOLO variants [R1–R3] for surface defect detection are not explicitly compared and discussed. Moreover, the motivation solely relies on a literature review of solar cell defect detection without contextualizing the problem within broader challenges. It fails to link existing methods’ limitations. It would be better to extensively review the latest YOLO variants.

Technology: The paper omits background theorems for integrating the crawfisher algorithm with MNS-YOLO. It would be better to explain how the algorithm’s optimization strategy enhances YOLO’s feature extraction. Additionally, some key components lack clear roles when describing technology. Furthermore, the paper is not well organized, some equations sound quite messy.

Experiment: According to its response, the authors not well emphasize and address each comment raised by the reviewers. The generalization ability should be evaluated on some commonly used datasets such as NEU-DET and GC10-DET.

[R1] Usamentiaga, R., Lema, D. G., Pedrayes, O. D., & Garcia, D. F. (2022). Automated surface defect detection in metals: a comparative review of object detection and semantic segmentation using deep learning. IEEE Transactions on Industry Applications, 58(3), 4203-4213.

[R2] X. Huang, J. Zhu and Y. Huo, SSA-YOLO: An Improved YOLO for Hot-Rolled Strip Steel Surface Defect Detection, in IEEE Transactions on Instrumentation and Measurement, vol. 73, pp. 1-17, 2024, Art no. 5040017

[R3] Cui, L., Jiang, X., Xu, M., Li, W., Lv, P., & Zhou, B. (2021). SDDNet: A fast and accurate network for surface defect detection. IEEE Transactions on Instrumentation and Measurement, 70, 1-13.

[R4] Tian Y, Ye Q, Doermann D. Yolov12: Attention-centric real-time object detectors[J]. arXiv preprint arXiv:2502.12524, 2025.

7. PLOS authors have the option to publish the peer review history of their article (what does this mean? ). If published, this will include your full peer review and any attached files.

**Do you want your identity to be public for this peer review?** For information about this choice, including consent withdrawal, please see our Privacy Policy .

Reviewer #2: No

Reviewer #3: No

Reviewer #4: No

---

## [Author Response · Author response to Decision Letter 2]

9 Aug 2025

Response to Reviewers

Fusion of crayfish optimization algorithm and MNS-YOLO for solar cell defect detection (ID: PONE-D-25-02298R1)

Dear Editor Prof. Xu,

On behalf of all the co-authors, I would like to thank you for providing us with the opportunity to revise the manuscript. Following your comments, we have carefully revised the manuscript. Changes in the revised manuscript are marked in red. The responses to the reviewers’ comments are detailed below.

Editor:

Thank you for your kind reminder regarding the journal’s requirements for the reference list. We have carefully reviewed all references to ensure their completeness and accuracy. Any necessary changes, including the replacement of outdated or non-English references with relevant current English-language sources, have been made and are clearly indicated in both the revised manuscript (highlighted in red) and in the rebuttal letter. We confirm that no retracted articles remain in the reference list without explicit justification, in full compliance with the journal’s publication standards.

Journal Requirements:

Response:Thank you for your professional comments. We have carefully considered your suggestions. We have thoroughly reviewed the entire reference list to ensure its completeness and accuracy. In the process, we found that Reference 8 is an EI-indexed paper written only in Chinese, without an SCI version. Therefore, we have replaced it with the latest relevant English-language literature and revised the corresponding content in the manuscript accordingly (Please see P3, in red).

In addition, we have standardized and unified the formatting of all references to meet the requirements of PLOS ONE. The newly added references and the updated Reference 8 have been highlighted in red in the revised manuscript. We have ensured that the updated reference list is complete, accurate, and free of any retracted citations that have not been clearly indicated, fully complying with the journal’s publication standards.

We appreciate your attention and guidance.

Reviewer #4:

The manuscript proposes CMNS-YOLO, a model combining the crawfisher optimization algorithm with MNS-YOLO for solar cell defect detection. However, critical flaws in motivation, theoretical foundation, and experimental design undermine the validity of the research. Some suggestions are as follows:

Motivation: Recent YOLO variants [R1–R3] for surface defect detection are not explicitly compared and discussed. Moreover, the motivation solely relies on a literature review of solar cell defect detection without contextualizing the problem within broader challenges. It fails to link existing methods’ limitations. It would be better to extensively review the latest YOLO variants.

Technology: The paper omits background theorems for integrating the crawfisher algorithm with MNS-YOLO. It would be better to explain how the algorithm’s optimization strategy enhances YOLO’s feature extraction. Additionally, some key components lack clear roles when describing technology. Furthermore, the paper is not well organized, some equations sound quite messy.

Experiment: According to its response, the authors not well emphasize and address each comment raised by the reviewers. The generalization ability should be evaluated on some commonly used datasets such as NEU-DET and GC10-DET.

[R1] Usamentiaga, R., Lema, D. G., Pedrayes, O. D., & Garcia, D. F. (2022). Automated surface defect detection in metals: a comparative review of object detection and semantic segmentation using deep learning. IEEE Transactions on Industry Applications, 58(3), 4203-4213.

[R2] X. Huang, J. Zhu and Y. Huo, SSA-YOLO: An Improved YOLO for Hot-Rolled Strip Steel Surface Defect Detection, in IEEE Transactions on Instrumentation and Measurement, vol. 73, pp. 1-17, 2024, Art no. 5040017

[R3] Cui, L., Jiang, X., Xu, M., Li, W., Lv, P., & Zhou, B. (2021). SDDNet: A fast and accurate network for surface defect detection. IEEE Transactions on Instrumentation and Measurement, 70, 1-13.

[R4] Tian Y, Ye Q, Doermann D. Yolov12: Attention-centric real-time object detectors[J]. arXiv preprint arXiv:2502.12524, 2025.

Response:Thank you for your professional comments. We have carefully considered your suggestions.

First, regarding the motivation, we have expanded the main text to include a broader and more comprehensive review of the latest YOLO models and their variants, highlighting the limitations of existing approaches (Please see P7, in red). This situates the problem within a wider context of challenges, beyond relying solely on literature related to solar cell defect detection, and also takes into account the detailed design of the model. In future work, we will further clarify and comparatively discuss recent YOLO variants applied to surface defect detection.

Second, regarding the technical aspects, we have introduced and explained the rationale for integrating the Crayfish Optimization Algorithm with MNS-YOLO (Please see P23, in red). We have described in detail the core concept of using the Crayfish Optimization Algorithm to optimize MNS-YOLO’s hyperparameters, thereby reducing its sensitivity to parameter settings, preventing it from falling into local optima during training, and further enhancing its feature extraction capability (Please see P24, in red).

In addition, to more clearly describe the role of certain key components, we have rewritten these descriptions using more professional terminology and an improved academic writing style (Please see P14, P19, in red). Furthermore, to make the equations more professional and easier to understand, we have refined their formatting (Please see Equations 2, 5–9, 10, 12–16, 18, 20–24, in red) and updated the corresponding symbol representations.

Finally, with respect to the experiments, we plan to evaluate the generalization ability of our method on several commonly used datasets (such as NEU-DET and GC10-DET) in future work.

We believe these revisions strengthen the broader significance of the study, make the logical chain more complete, and enhance the technical soundness and credibility of our work. We appreciate your suggestions, which have helped us improve the quality of the manuscript and better convey its research value to readers in the field, providing a more meaningful reference for solar cell defect detection.

---

## [Decision Letter · Decision Letter 2]

13 Aug 2025

PONE-D-25-02298R2Fusion of crayfish optimization algorithm and MNS-YOLO for solar cell defect detectionPLOS ONE

Dear Dr. zhang,

Thank you for submitting your manuscript to PLOS ONE. After careful consideration, we feel that it has merit but does not fully meet PLOS ONE’s publication criteria as it currently stands. Therefore, we invite you to submit a revised version of the manuscript that addresses the points raised during the review process.

We look forward to receiving your revised manuscript.

Kind regards,

Xu Yanwu

Academic Editor

PLOS ONE

**Journal Requirements:**

**Additional Editor Comments:**

Please fix the problems from Reviewer 4.

Reviewers' comments:

Reviewer's Responses to Questions

**Comments to the Author**

1. If the authors have adequately addressed your comments raised in a previous round of review and you feel that this manuscript is now acceptable for publication, you may indicate that here to bypass the “Comments to the Author” section, enter your conflict of interest statement in the “Confidential to Editor” section, and submit your "Accept" recommendation.

Reviewer #2: All comments have been addressed

Reviewer #3: All comments have been addressed

Reviewer #4: All comments have been addressed

2. Is the manuscript technically sound, and do the data support the conclusions?

Reviewer #2: Yes

Reviewer #3: Yes

Reviewer #4: Yes

3. Has the statistical analysis been performed appropriately and rigorously? 

Reviewer #2: Yes

Reviewer #3: Yes

Reviewer #4: I Don't Know

4. Have the authors made all data underlying the findings in their manuscript fully available?

Reviewer #2: Yes

Reviewer #3: Yes

Reviewer #4: No

5. Is the manuscript presented in an intelligible fashion and written in standard English?

Reviewer #2: Yes

Reviewer #3: Yes

Reviewer #4: No

6. Review Comments to the Author

**Reviewer #2: ** (No Response)

**Reviewer #3:**  Thank you very much for your consideration of my comments which had been addressed well, so I have no concern for publication

**Reviewer #4: ** Some equations should be formulated, such as Eqn. 25-28. As well, some terms are not well explained.

Regarding experimental environment: The experiments were conducted on PyTorch or Matlab?

It is recommended to add the explanation of A, B, C, D in the table captain (Table 3).

Most of references misuse "First Name" and "Last Name". Please check them all.

7. PLOS authors have the option to publish the peer review history of their article (what does this mean? ). If published, this will include your full peer review and any attached files.

**Do you want your identity to be public for this peer review?** For information about this choice, including consent withdrawal, please see our Privacy Policy .

Reviewer #2: No

Reviewer #3: No

Reviewer #4: No

---

## [Author Response · Author response to Decision Letter 3]

17 Sep 2025

Response to Reviewers

Fusion of crayfish optimization algorithm and MNS-YOLO for solar cell defect detection (ID: PONE-D-25-02298R1)

Dear Editor Prof. Xu,

On behalf of all the co-authors, I would like to thank you for providing us with the opportunity to revise the manuscript. Following your comments, we have carefully revised the manuscript. Changes in the revised manuscript are marked in red. The responses to the reviewers’ comments are detailed below.

Editor:

Thank you for your kind reminder regarding the journal’s requirements for the reference list. We have carefully reviewed all references to ensure their completeness and accuracy. Any necessary changes, including the replacement of outdated or non-English references with relevant current English-language sources, have been made and are clearly indicated in both the revised manuscript (highlighted in red) and in the rebuttal letter. We confirm that no retracted articles remain in the reference list without explicit justification, in full compliance with the journal’s publication standards.

Journal Requirements:

Response: Thank you for your professional comment. We have considered your opinion. We have thoroughly and comprehensively reviewed and evaluated the reviewer's comments (including suggestions, references to previously published works), and determined that they are highly relevant to the content of this article.

Response:Thank you for your professional comment. We have considered your opinion. We have thoroughly checked all reference lists to ensure their completeness and accuracy. In addition, we have checked and revised all the references raised by Reviewer 4 for any issues. In addition, all reference formats have been standardized and adjusted to comply with the requirements of PLOS ONE journal. The revised literature has been marked in red in the revised manuscript. We have ensured that the revised reference list is complete, accurate, and free of unexplained retracted references, meeting the publication requirements of the journal. Thank you for your attention and guidance.

Reviewer #4:

Some equations should be formulated, such as Eqn. 25-28. As well, some terms are not well explained.

Regarding experimental environment: The experiments were conducted on PyTorch or Matlab?

It is recommended to add the explanation of A, B, C, D in the table captain (Table 3).

Most of references misuse "First Name" and "Last Name". Please check them all.

Response:Thank you for your professional comment. We have considered your feedback and made the following modifications:

Firstly, we have checked all the formulas, carefully examined each parameter in the formulas, and added detailed terminology explanations for each parameter to improve the readability of the article and ensure that readers can accurately understand the meaning and function of each parameter.(Please see P17-P19) Secondly, regarding the experimental environment, we have revised the wording to clarify that all experiments were conducted using Python 3.8 and Pytorch 1.10.0 in Jupyter Notebook.(Please see P27) This modification clarifies the specific experimental environment, making it easier for readers to reproduce our research results. Then, we added explanations for A, B, C, and D in the table length of Table 3 to further help readers understand the different experimental groups, comparison conditions, and evaluation indicators represented by these labels, making the logic and content of Table 3 more complete.(Please see P32) In addition, we carefully examined the paper and made corresponding modifications to make its expression clearer and content more complete. (Please see P16,P30)Finally, we thoroughly examined and corrected the issue of improper use of names and surnames in the references to ensure that the citation format of author names conforms to academic standards, avoiding confusion caused by inconsistent naming, and enhancing the academic rigor of the reference section.(Please see P41-P52)

We would like to thank you again for your professional advice. We believe that through these revisions, not only has the quality of the paper been improved, but the accuracy of terminology expression has also been enhanced, and the research value can be clearly conveyed to readers in the field, providing a reference solution for the field of solar cell defect detection.

---

## [Editor Report · Decision Letter 3]

22 Sep 2025

Fusion of crayfish optimization algorithm and MNS-YOLO for solar cell defect detection

PONE-D-25-02298R3

Dear Dr. zhang,

We’re pleased to inform you that your manuscript has been judged scientifically suitable for publication and will be formally accepted for publication once it meets all outstanding technical requirements.

Kind regards,

Xu Yanwu

Academic Editor

PLOS ONE
---

## [Editor Report · Acceptance letter]

PONE-D-25-02298R3

PLOS ONE

Dear Dr. zhang,

I'm pleased to inform you that your manuscript has been deemed suitable for publication in PLOS ONE. Congratulations! Your manuscript is now being handed over to our production team.

Kind regards,

on behalf of

Dr. Xu Yanwu

Academic Editor

PLOS ONE